# Disentangling the multiorbital contributions of excitons by photoemission exciton tomography

Wiebke Bennecke [1], Andreas Windischbacher [2], David Schmitt[1], Jan Philipp Bange [1], Ralf Hemm[3], Christian S. Kern [2], Gabriele D'Avino [4], Xavier Blase [4], Daniel Steil [1], Sabine Steil[1], Martin Aeschlimann [3], Benjamin Stadtmüller [3], Marcel Reutzel [1], Peter Puschnig [2], G. S. Matthijs Jansen [1] ✉ & Stefan Mathias [1,5] ✉

Excitons are realizations of a correlated many-particle wave function, specifically consisting of electrons and holes in an entangled state. Excitons occur widely in semiconductors and are dominant excitations in semiconducting organic and low-dimensional quantum materials. To efficiently harness the strong optical response and high tuneability of excitons in optoelectronics and in energy-transformation processes, access to the full wavefunction of the entangled state is critical, but has so far not been feasible. Here, we show how time-resolved photoemission momentum microscopy can be used to gain access to the entangled wavefunction and to unravel the exciton's multiorbital electron and hole contributions. For the prototypical organic semiconductor buckminsterfullerene ($C_{60}$), we exemplify the capabilities of exciton tomography and achieve unprecedented access to key properties of the entangled exciton state including localization, charge-transfer character, and ultrafast exciton formation and relaxation dynamics.

Optical excitations in semiconducting materials deposit energy that can, in the best case, be harnessed in optoelectronic and photovoltaic devices. This potential for energy harvesting holds true over an extremely wide range of semiconducting materials, extending from classical silicon to two-dimensional transition metal dichalcogenides, perovskites and organic semiconductors[1–4]. Hence, major experimental and theoretical research efforts strive to understand such optical excitations.

On the fundamental level, the primary response to the optical excitation is excitonic: Coulomb-correlated electron-hole pairs are created. In the most simple picture, for an organic semiconductor, such an excitation can be understood by the simultaneous creation of

an excess electron in the lowest unoccupied molecular orbital (LUMO) and an excess hole in the highest occupied molecular orbital (HOMO). A very fundamental manifestation of the correlated interaction between the electron and hole is the exciton binding energy, which can be observed in optical absorption spectroscopy from the appearance of an absorption feature at one exciton binding energy below the single-particle band gap[5]. Hence, the correlation of the many-body wavefunction serves to reduce the required energy to place a single electron into an excited state.

The concept of excitons does not stop at the lowest HOMO-LUMO excitations, and it provides a natural description of higher excitations as well. This includes transitions that may be described by an electron

[1]I. Physikalisches Institut, Georg-August-Universität Göttingen, Friedrich-Hund-Platz 1, 37077 Göttingen, Germany. [2]Institute of Physics, University of Graz, NAWI Graz, Universitätsplatz 5, 8010 Graz, Austria. [3]Department of Physics and Research Center OPTIMAS, University of Kaiserslautern-Landau, Erwin-Schrödinger-Straße 46, 67663 Kaiserslautern, Germany. [4]Univ. Grenoble Alpes, CNRS, Inst NEEL, F-38042 Grenoble, France. [5]International Center for Advanced Studies of Energy Conversion (ICASEC), University of Göttingen, Göttingen, Germany. ✉e-mail: gsmjansen@uni-goettingen.de; smathias@uni-goettingen.de

in a higher conduction level or a hole in a lower valence level, or the build-up of trions and biexcitons that consist of three or four entangled charged particles[6,7]. Usually, the exciton wavefunction $\psi_m$ is described (in the Tamm-Dancoff approximation) by a superposition of multiple electron-hole pairs[8]:

$$\psi_m(\boldsymbol{r}_h, \boldsymbol{r}_e) = \sum_{v,c} X_{vc}^{(m)} \phi_v^*(\boldsymbol{r}_h)\chi_c(\boldsymbol{r}_e). \tag{1}$$

Here $\phi_v$ and $\chi_c$ are the $v^{th}$ valence and $c^{th}$ conduction states of the ground-state system, respectively. The coefficients $X_{vc}^{(m)}$ can create an entangled state where the electron and hole coordinates ($\boldsymbol{r}_e$ and $\boldsymbol{r}_h$) cannot be considered independently (Fig. 1). Access to this orbital picture of the excitonic wavefunction is highly valuable[9], because imaging of the full entangled state would give direct access to exciton properties such as localization and charge-transfer character. Indeed, this information is particularly critical in the case of organic semiconductors, where it is well-known that such multiorbital correlated quasiparticles dominate the energy landscape[10,11]. However, it must be emphasized that conventional optical spectroscopy methods, including absorption and fluorescence spectroscopy, only provide access to the exciton energy $\Omega_m$, and do not provide any information about the multiorbital contributions ($\phi_v^*\chi_c$) of the exciton (see Fig. 1a, b).

For single-particle molecular orbitals in organic semiconductors, an imaging of the wavefunction is possible through photoemission orbital tomography[12,13]. In the last years, this technique was increasingly used to study light-induced dynamics in organic semiconductors[14,15] and 2D quantum materials[16–18], which exemplified the tremendous capabilities of this technique when applied to excitonic states. However, the full potential of photoemission exciton tomography was only recently indicated in a theoretical study by Kern et al., and promises to unravel the entangled single-particle orbital contributions and real-space properties of the excitons[19]. In this article, we experimentally introduce photoemission exciton tomography (Fig. 1c) and use it for the first time to characterize the correlated excitonic electron-hole state in an organic semiconductor. Specifically, we find that the detected photoelectron kinetic energy provides a sensitive probe of the hole component of the exciton, and furthermore that the photoelectron momentum map probes the spatial properties of the electron component.

## Results and discussion

### The exciton spectrum of $C_{60}$

In order to introduce the photoemission exciton tomography approach, we select $(C_{60} - I_h)[5,6]$fullerene ($C_{60}$) as an ideal, widely used[20–22] and prototypical example. In particular, $C_{60}$ shows a series of optical absorption features in multilayer and other aggregated structures[23], where different spectroscopy studies have indicated that these optical transitions correspond to the formation of excitons with differing charge-transfer character[24–28]. Although these indirect results are supported by time-dependent density functional theory calculations[29–31], quantitative access to the multiorbital wavefunction contributions (see Eq. (1) and Fig. 1) has so far not been feasible. Thus, the $C_{60}$ organic semiconductor is an ideal platform to showcase the capabilities of photoemission exciton tomography.

In theory, we obtain the exciton spectrum by employing the many-body framework of $GW$ and Bethe-Salpeter-Equation ($GW$+BSE) calculations on top of a hybrid-functional density functional theory (DFT) ground state calculation (Fig. 2a, see Methods)[8,32]. We find that the $C_{60}$ crystal (Fig. 2a) can be accurately described using two symmetry-inequivalent $C_{60}$ dimers (see Supplementary Information, Supplementary Figs. S1, S4 and S5 for an experimental determination of the crystal structure and a convergence analysis of the dimer model). The calculated single-particle energy levels are shown in Fig. 2b, where we group the electron removal and electron addition energies into four bands, denoted according to the parent orbitals of the gas-phase $C_{60}$ molecule. Building upon the $GW$ single-particle energies (Fig. 2b), we solve the Bethe-Salpeter equation and compute the energies $\Omega_m$ of all correlated electron-hole pairs (excitons), which results in the absorption spectrum shown in the bottom panel of Fig. 2c. Furthermore, we obtain the weights $X_{vc}^{(m)}$ on the specific electron-hole pairs that build-up the $m^{th}$ exciton state by Eq. (1). This provides a full description of the multiorbital entangled excitons, and thereby contains the full spatial properties of each exciton.

To gain more insight into the character of the excitons $\psi_m$, we qualitatively classify them according to the most dominant orbital contributions that are involved in the transitions. This is visualized in the four sub-panels above the absorption spectrum in Fig. 2c. For a given exciton energy $\Omega_m$, the black bars in each sub-panel show the partial contribution $|X_{vc}^{(m)}|^2$ of characteristic electron-hole transitions $\phi_v\chi_c$ to a given exciton $\psi_m$. Looking at individual sub-panels, we see

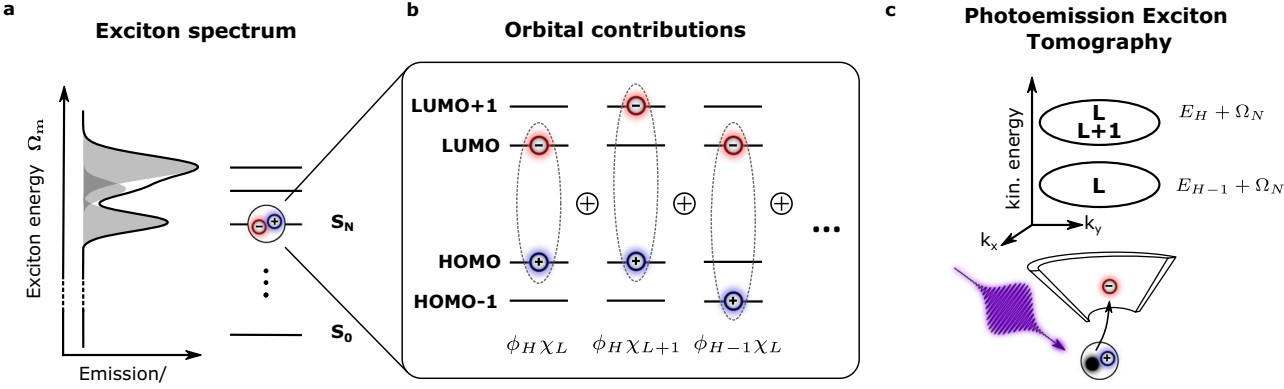

**Fig. 1 | The optical response of many semiconducting materials is described by the excitation of excitons. a** In the exciton picture, each excited state is described by an exciton energy $\Omega_m$, which (for bright excitons) can be measured by optical spectroscopy. **b** At the orbital level, however, each exciton is built up by an entangled sum of electron-hole pairs $\phi_v\chi_c$ (see Eq. (1)). In this description, the sum of orbital contributions provides complete access to the spatial properties of the exciton. **c** In photoemission exciton tomography, a high-energy photon photo-emits the electron and thereby breaks up the exciton. The single-particle electron orbitals (here LUMO (L) and LUMO+1 (L+1)) contributing to the exciton are imprinted on the photoelectron momentum distribution, while the kinetic energy distribution probes the contributing hole orbitals by measuring the hole binding energy $E_H$ (see Eq. (2)). A full momentum- and energy-resolved measurement of the photoelectron spectrum therefore provides an ideal starting point for a comparison to ab-initio calculations of the excitonic wavefunction, and thereby provides access to the spatial properties of the exciton.

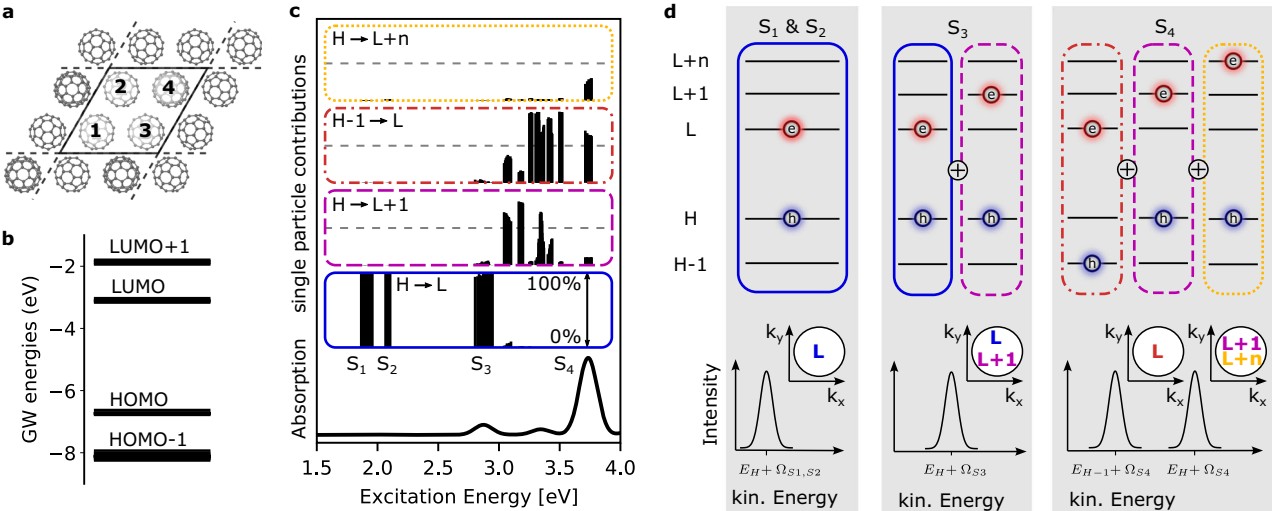

**Fig. 2 | Ab-initio calculation of the electronic structure and exciton spectrum of C$_{60}$ dimers in a crystalline multilayer sample. a** The unit cell for a monolayer of C$_{60}$, for which *GW*+BSE calculations for the dimers 1-2 and 3-4 were performed. **b** Electron addition/removal single-particle energies as retrieved from the self-consistent *GW* calculation. These energies directly provide $\varepsilon_v$ in Eq. (2). The HOMO-LUMO manifolds consist of 18, 10, 6, and 6 energy levels per dimer, respectively, originating from the $g_g+h_g$, $h_u$, $t_{1u}$, and $t_{1g}$ irreducible representations of the gas phase C$_{60}$ orbitals[56]. **c** Results of the full *GW*+BSE calculation in accordance with

ref. 23, showing as a function of the exciton energy $\Omega$ from bottom to top: the calculated optical absorption, the exciton band assignment S$_1$ - S$_4$, and the relative contributions to the exciton wavefunctions of different electron-hole pair excitations $\phi_v\chi_c$. Full details on the calculations are given in the Methods section. **d** Sketch of the composition of the exciton wavefunction of the S$_1$ - S$_4$ bands and their expected photoemission signatures based on Eq. (2). In order to visualize the contributing orbitals, blue holes and red electrons are assigned to the single-particle states as shown in (**b**).

that not only the excitons are commonly composed of multiple characteristic electron-hole transitions, but also that single electron-hole transitions can belong to different excitons $\psi_m$ that have very different exciton energies $\Omega_m$. For example, the blue panel in Fig. 2c shows the contributions of HOMO → LUMO (abbreviated H → L) transitions as a function of exciton energy $\Omega_m$, and we see that these transitions contribute to excitons that are spread in energy over a scale of more than 1 eV (from $\Omega_m \approx 1.7$ eV–3 eV). This spread of H → L contributions (and also H − n → L+m contributions) is caused by the fact that there are already many orbital energies per dimer (see Fig. 2b) which combine to form excitons with different degrees of localization and delocalization of the electrons and holes on one or more molecules.

We now focus on four exciton bands of the C$_{60}$ film, denoted as S$_1$ - S$_4$, which are centered around $\Omega_{S1}$, $\Omega_{S2}$, $\Omega_{S3}$ and $\Omega_{S4}$ at 1.9, 2.1, 2.8 and 3.6 eV, respectively. Notably, S$_2$ and S$_3$ were previously found to have charge-transfer character[24,25], while S$_4$ stands out due to a fundamentally different wavefunction composition. It is important to emphasize that each exciton band S$_1$ - S$_4$ arises from many individual excitons $\psi_m$ with similar exciton energies $\Omega_m$ within the exciton band. From Fig. 2c, we see that the S$_1$ and S$_2$ exciton bands are made up of excitons $\psi_m$ that are almost exclusively composed of transitions from H → L. On the other hand, S$_3$ shows in addition to H → L also weak contributions from H → L+1 transitions (pink-dashed panel). The S$_4$ exciton band can be characterized as arising from H → L+1 (pink-dashed panel) and H − 1 → L (orange-dash-dotted panel) as well as transitions from the HOMO to several higher lying orbitals denoted as H → L+n (yellow-dotted panel). Note that the inclusion of electron-hole correlations has important consequences on the composition of the exciton wave function $\psi_m$[33,34]. Specifically, S$_3$ is not only composed of H → L transitions but exhibits also an admixture of H → L+1 transitions despite the calculated ≈1 eV energy separation of quasiparticle LUMO and LUMO+1 levels.

## Photoemission signature of multiorbital entangled excitons
In the following, we investigate whether these theoretically predicted multiorbital characteristics of the excitons can also be probed

experimentally. Therefore, we take the exciton of Eq. (1) as the initial state and apply the common plane-wave final state approximation of photoemission orbital tomography[19]. The photoemission intensity of the exciton $\psi_m$ is formulated as

$$I_m(E_{\text{kin}}, \boldsymbol{k}) \propto |\boldsymbol{A}\boldsymbol{k}|^2 \sum_v \left| \sum_c X_{vc}^{(m)} \mathcal{F}[\chi_c](\boldsymbol{k}) \right|^2 \times \delta(h\nu - E_{\text{kin}} - \varepsilon_v + \Omega_m). \tag{2}$$

Here $\boldsymbol{A}$ is the vector potential of the incident light field, $\mathcal{F}$ the Fourier transform, $\boldsymbol{k}$ the photoelectron momentum, $h\nu$ the probe photon energy, $\varepsilon_v$ the $v^{\text{th}}$ ionization potential, $\Omega_m$ the exciton energy, and $E_{\text{kin}}$ the energy of the photoemitted electron. Note that $\varepsilon_v$ directly indicates the final-state energy of the left-behind hole. In the context of our present study, delving into Eq. (2) leads to two striking consequences that allow us to disentangle the electron and hole contributions of the exciton, which we discuss in the following.

First, we will discuss the importance of the hole contribution based on the consequences of the multiorbital entangled character for the photoelectron spectrum for the four different exciton bands in C$_{60}$. In Fig. 2d, we sketch the typical single-particle energy level diagrams for the HOMO and LUMO states and indicate the contributing orbitals to the two-particle exciton state by blue holes and red electrons in these states, respectively. For the S$_1$ exciton band (left panel), we already found that the main orbital contributions to the band are of H → L character (Fig. 2d, left, and see Fig. 2c, blue panel). To determine the kinetic energy of the photoelectrons originating from the exciton, we have to consider the correlated nature of the electron-hole pair. The energy conservation expressed by the delta function in Eq. (2) (see also refs. 35–37) requires that the kinetic energy of the photoelectron depends on the ionization energy of the involved HOMO hole state $\varepsilon_v = \varepsilon_H$ and the correlated electron-hole pair energy $\Omega \approx \Omega_{S1}$. Therefore, we expect to measure a single photoelectron peak, as shown in the lower part of the left panel of Fig. 2d. In the case of the S$_2$ exciton the situation is similar, since the main orbital contributions are also of H → L character. However, since the S$_2$ exciton band has a higher

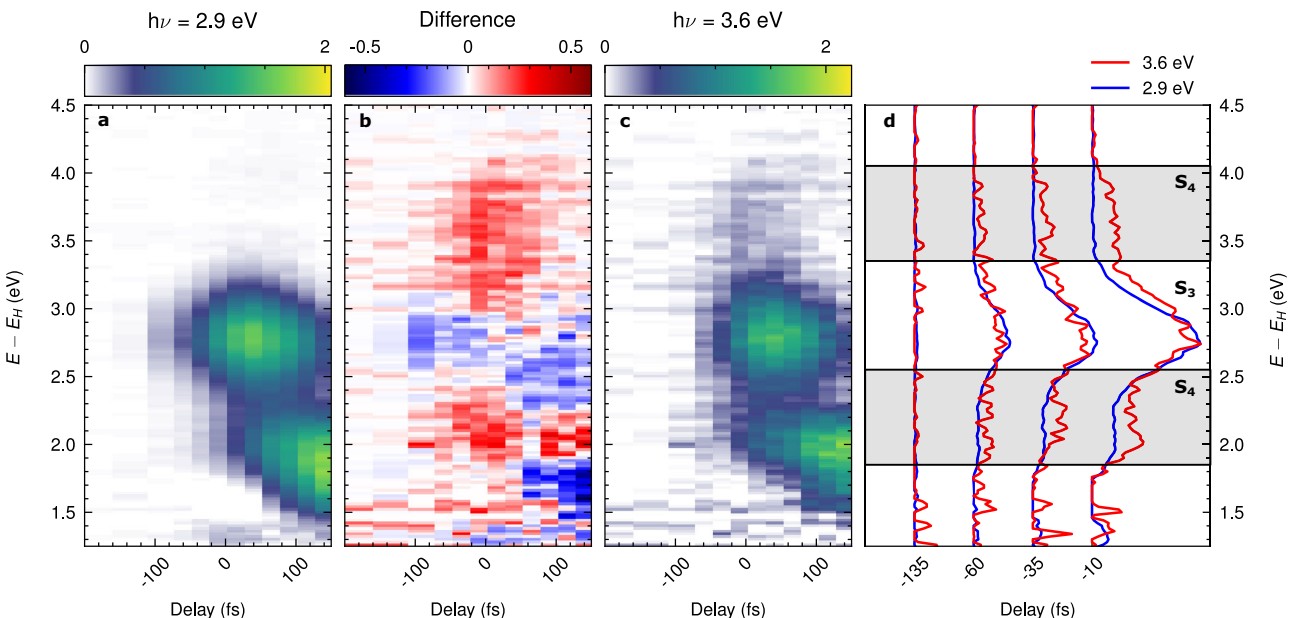

**Fig. 3 | Comparison of the exciton dynamics of multilayer $C_{60}$ for hν = 2.9 eV excitation and hν = 3.6 eV excitation. a**, **c** show the time-resolved photoelectron spectra, both normalized and shifted in time to match the intensity of the $S_3$ signals (see Methods for full details on the data analysis). As can be seen in the difference **b** for hν = 3.6 eV pump we observe an enhancement of the photoemission yield around $E − E_H ≈ 3.6$ eV as well as around $E − E_H ≈ 2.2$ eV. We attribute this signal to the $S_4$ exciton band, which has hole contributions stemming from both the HOMO and the HOMO-1. To further quantify the signal of the $S_4$ exciton, **d** shows energy distribution curves for both measurements at early delays, showing the enhancement in the hν = 3.6 eV measurement.

energy $\Omega_{S2}$, the photoelectron peak is also located at a higher kinetic energy with respect to the $S_1$ peak.

In the case of the $S_3$ exciton band, we find that in contrast to the $S_1$ and $S_2$ excitons not only H → L, but also H → L+1 transitions contribute (Fig. 2d, middle panel, and see Fig. 2c, blue and pink-dashed panels, respectively). However, we still expect a single peak in the photoemission, because the same hole states are involved for both transitions (i.e., same $\varepsilon_v = \varepsilon_H$ in the sum in Eq. (2)), and all orbital contributions have the same exciton energy $≈\Omega_{S3}$, even though transitions with electrons in energetically very different single-particle LUMO and LUMO+1 states contribute. With other words, and somewhat counter-intuitively, the single-particle energies of the electron orbitals (the LUMOs) contributing to the exciton do not enter the energy conservation term in Eq. (2), and thus do not affect the kinetic energy observed in the experiment.

Finally, for the $S_4$ exciton band at $\Omega_{S4} = 3.6$ eV, we find three major contributions (Fig. 2d, right panel), where not only the electrons but also the holes are distributed over two energetically different levels, namely the HOMO (see pink-dashed and yellow-dotted panels in Fig. 2c, d) and the HOMO-1 (see orange-dash-dotted panels in Fig. 2c, d). Thus, there are two different final states available for the hole, each with a different binding energy. Consequently, the photoemission spectrum of $S_4$ is expected to exhibit a double-peak structure with intensity appearing $≈3.6$ eV above the HOMO kinetic energy $E_H$, and $≈3.6$ eV above the HOMO-1 kinetic energy $E_{H−1}$, as illustrated in the right-most panel of Fig. 2d. Relating this specifically to the single-particle picture of our GW calculations, the two peaks are predicted to have a separation of $\varepsilon_{H−1} − \varepsilon_H = (8.1 − 6.7)$ eV $= 1.4$ eV. In summary, the photoelectron energy distribution provides access to the multiorbital character of the exciton, because different hole states that contribute to the exciton induce a multipeak structure in the spectrum.

The second consequence of Eq. (2) concerns the electron contribution, which modifies the photoemission momentum distribution. In analogy to conventional photoemission orbital tomography, Eq. (2) provides the theoretical framework for interpreting time- and

momentum-resolved data from excitons. Ground state momentum maps can be easily understood in terms of the Fourier transform $\mathcal{F}$ of single-particle orbitals[12]. A naive extension to excitons might imply an incoherent, weighted sum of all conduction orbitals $\chi_c$ contributing to the exciton wavefunction. However, as Eq. (2) shows, such a simple picture proves insufficient. Instead, the momentum pattern of the exciton wavefunction is related to a coherent superposition of the electron orbitals $\chi_c$ weighted by the electron-hole coupling coefficients $X_{vc}^{(m)}$. The implications of this finding are sketched in the $k_x$-$k_y$ plots in Fig. 2d and are most obvious for the $S_3$ band. Here, the exciton is composed of transitions with a common hole position, i.e., H → L and H → L+1, leading to a coherent superposition of all 12 electron orbitals from the LUMO and LUMO+1 in the momentum distribution. In summary, multiple hole contributions can be identified in a multi-peak structure in the photoemission spectrum, and multiple electron contributions will result in a coherent sum of the electron orbitals that can be identified in the corresponding energy-momentum patterns of time-resolved data.

## Disentangling multiorbital contributions experimentally

These very strong predictions about multi-peaked photoemission spectra due to multiorbital entangled excitons can be directly verified in an experiment on $C_{60}$ by comparing spectra for resonant excitation of either the $S_3$ or the $S_4$ excitons (see Fig. 2). We employ our recently developed setup for photoelectron momentum microscopy[38,39] and use ultrashort laser pulses to optically excite the $S_3$ and the $S_4$ bright excitons in $C_{60}$ thin films that were deposited on Cu(111) (measurement temperature T ≈ 80 K, p-polarized excitation; see Methods). The corresponding time-resolved photoelectron spectra of the electrons that were initially part of the bound electron-hole pairs are shown in Fig. 3a and c, respectively. Starting from the excitation of the $S_3$ exciton band with hν = 2.9 eV photon energy (which is sufficiently resonant to excite the manifold of exciton states that make up the $S_3$ band around $\Omega_{S3} = 2.8$ eV), we can clearly identify the direct excitation (at 0 fs delay) of the exciton $S_3$ feature at an energy of $E ≈ 2.8$ eV above the kinetic energy $E_H$ of the HOMO level. Shortly after the excitation, additional

photoemission intensity builds up at $E - E_H \approx 2.0$ eV and $\approx 1.7$ eV, which is known to be caused by relaxation to the $S_2$ and $S_1$ dark exciton states[25] and is in good agreement with the theoretically predicted energies of $E - E_H \approx 2.1$ eV and $\approx 1.9$ eV (see Fig. 2c, blue panel).

Changing now the pump photon energy to $h\nu = 3.6$ eV for direct excitation of the $S_4$ exciton band (Fig. 3c), two distinct peaks at $\approx 3.6$ eV above the HOMO and $\approx 3.6$ eV above the HOMO-1 are expected from theory. While photoemission intensity at $E - E_H \approx 3.6$ eV above the HOMO level is readily visible in Fig. 3c, the second feature at 3.6 eV above the HOMO-1 is expected at $E - E_H \approx 2.2$ eV above the HOMO level (corresponding to $E - E_{H-1} \approx 3.6$ eV) and thus almost degenerate with the aforementioned $S_2$ dark exciton band at about $E - E_H \approx 2.0$ eV, which appears after the optical excitation due to relaxation processes. We note that a fast relaxation to the band of exciton states between 3.0 and 3.5 eV (see Fig. 2c) is also possible and may contribute to the observed signal. However, these excitons also have contributions from the HOMO and HOMO-1 and are thus also predicted to lead to two distinct peaks at somewhat lower $\approx 3.4$ eV above the HOMO and HOMO-1, thereby also confirming our expectations from theory. In any case, we therefore need to pinpoint the second $H - 1 \rightarrow L$ lower energy contribution from either the $S_4$ or the exciton band around 3.0–3.5 eV, and we do this by analyzing our data at the earliest time of excitation, i.e. before relaxation to the $S_2$ dark exciton band occurs, which is degenerate at this photoelectron energy of $\approx 2.2$ eV. Indeed, we find that a closer look around 0 fs delay shows additional photoemission intensity at this particular energy. Using a difference map (Fig. 3b) and direct comparisons of energy-distribution-curves at selected time-steps (Fig. 3d), we clearly find a double-peak structure corresponding to the energy difference of $\approx 1.4$ eV of the HOMO and HOMO-1 levels. Thereby, we have shown that photoelectron spectroscopy, in contrast to other techniques (e.g., absorption spectroscopy), is indeed able to disentangle different orbital contributions of the excitons. In this way, we have validated the theoretically predicted multi-peak structure of the multiorbital exciton state that is implied by Eq. (2). We also see that the photoelectron energies in the spectrum turn out to be sensitive probes of the corresponding hole contributions of the correlated exciton states.

We note that the signature of the $S_3$ excitons, even if not directly excited with the light pulse in this measurement, is still visible and moreover with significantly higher intensity than the multiorbital signals of the resonantly excited $S_4$ exciton band. The explanation for this effect is two-fold: first, the time-resolved signature suggests a very fast relaxation of the $S_4$ excitons to the $S_3$, with relaxation times well below 50 fs (see Supplementary Fig. S3). Second, a calculation following Eq. (2) predicts an about threefold reduced photoemission matrix element for the $S_4$ compared to the $S_3$ band, explaining the overall weaker signal.

## Time-resolved photoemission exciton tomography

In the full time-resolved photoemission experiment, by following the time-evolution of all electrons that were initially part of bound electron-hole pairs, one can observe how the optically-excited states relax to energetically lower-lying dark exciton states[17,18,24,25,35,40,41] (Supplementary Fig. S3) with the expected different localization and charge-transfer character. Importantly, the photoemission momentum microscope collects full time- and momentum-resolved data (i.e., 4D data set with time, energy and 2D momentum resolution), which we now show is ideal to access the spatial properties of the entangled multiorbital contributions.

We once again excite the $S_3$ exciton band in the $C_{60}$ film with $h\nu = 2.9$ eV pump energy, and collect the momentum fingerprints of the directly excited $S_3$ excitons around 0 fs and the subsequently built-up dark $S_2$ and $S_1$ excitons that appear in the exciton relaxation cascade in the $C_{60}$ film (see Fig. 4a, where the momentum maps of the lowest energy $S_1$ exciton band, the $S_2$ and the highest energy $S_3$ exciton band

are plotted from left to right; see Supplementary Fig. S3 for time-resolved traces of the exciton formation and relaxation dynamics). We note that the collection of the $S_1$, $S_2$, and $S_3$ momentum maps already required integration times of up to 70 hours and a summation of the data over all measured time-steps from -200 fs–15 ps (see Methods), so that a measurement of the comparatively low-intensity $S_4$ feature when excited with $h\nu = 3.6$ eV has not yet proved feasible. For the interpretation of the collected momentum maps from the $S_1$, $S_2$, and $S_3$ excitons, we also calculate the expected momentum fingerprints for the wavefunctions obtained from the $GW$+BSE calculation for both dimers, each rotated to all occurring orientations in the crystal. Finally, for the theoretical momentum maps, we sum up the photoelectron intensities of each electron-hole transition in an energy range of 200 meV centered on the exciton band. The results are shown in Fig. 4c below the experimental data for direct comparison.

First, recapitulating from Fig. 2c that the $S_1$ and the $S_2$ exciton bands are both only comprised of $H \rightarrow L$ transitions, we expect nearly identical momentum maps (but at different energies). Indeed, the experimental $S_1$ and $S_2$ momentum maps are largely similar (Fig. 4a), showing six lobes centered at $k_\parallel \approx 1.25$ Å$^{-1}$. These six-lobe features, as well as the energy splitting between $S_1$ and $S_2$ (see Fig. 2c), are accurately reproduced by the $GW$+BSE prediction (Fig. 4b). Furthermore, the $GW$+BSE calculation also shows identical momentum maps for $S_1$ and $S_2$ confirming that the result of the coherent sum over all $H \rightarrow L$ transitions (Eq. (2)) is similar, and also indicates that the $S_1$ and the $S_2$ exhibit a similar spatial structure of the exciton wavefunction. This is in contrast to a naive application of static photoemission orbital tomography to the unoccupied orbitals of the DFT ground state of $C_{60}$, which does indicate a similar momentum map for the LUMO, but cannot explain a kinetic energy difference in the photoemission signal, nor give any indication of differences in the corresponding exciton wavefunctions. With this agreement between experiment and theory, we now extract the spatial properties of the $GW$+BSE exciton wavefunctions. To visualize the degree of charge-transfer of these two-particle exciton wavefunctions $\psi_m(\mathbf{r}_h, \mathbf{r}_e)$, we integrate the electron probability density over all possible hole positions $\mathbf{r}_h$, considering only hole positions at one of the $C_{60}$ molecules in the dimer. This effectively fixes the hole contribution to a particular $C_{60}$ molecule (blue circles in Fig. 4c indicate the boundary of considered hole positions around one molecule, hole distribution not shown), and provides a probability density for the electronic part of the exciton wavefunction in the dimer, which we visualize by a yellow isosurface (see Fig. 4c). For the $S_1$ and $S_2$, when the hole position is restricted to one molecule of the dimer, the electronic part of the exciton wavefunction is localized at the same molecule of the dimer. Our calculations thus suggest that the $S_1$ and $S_2$ excitons are of Frenkel-like nature. Their energy difference originates from different excitation symmetries possible for the $H \rightarrow L$ transition (namely $t_{1g}$, $t_{2g}$, and $g_g$ for the $S_1$ and $h_g$ for the $S_2$)[31].

Compared to the $S_1$ and $S_2$ exciton bands, the $S_3$ band is not only composed of $H \rightarrow L$ transitions, but also has a minor contribution of $H \rightarrow L+1$ transitions (see Fig. 2c). We therefore expect that the $S_3$ momentum map cannot be identical to the $S_1$ and $S_2$ momentum maps, but must show a signature of the $H \rightarrow L+1$ contribution in addition to a possibly different coherent sum of all involved $H \rightarrow L$ transitions. Indeed, a closer look at the experimental data shows a more spoke-like structure for $S_3$, which is marked with red arrows for three of the six spokes in the raw data as a guide to the eye (Fig. 4a, top right). An analysis over all different orientations, i.e. effectively symmetrizing the data, makes the spoke-structure even better visible (Fig. 4a, bottom right, and Supplementary Fig. S6 for momentum lineouts). Hence, our experimental data clearly confirms the different character of the $S_3$ exciton band in comparison to $S_1$ and $S_2$. Looking at the theoretical data, we also find differences between the nearly identical $S_1$ and $S_2$ momentum maps (Fig. 4b) in comparison to the $S_3$ momentum map. Once again, the differences are marked with red arrows as a guide to

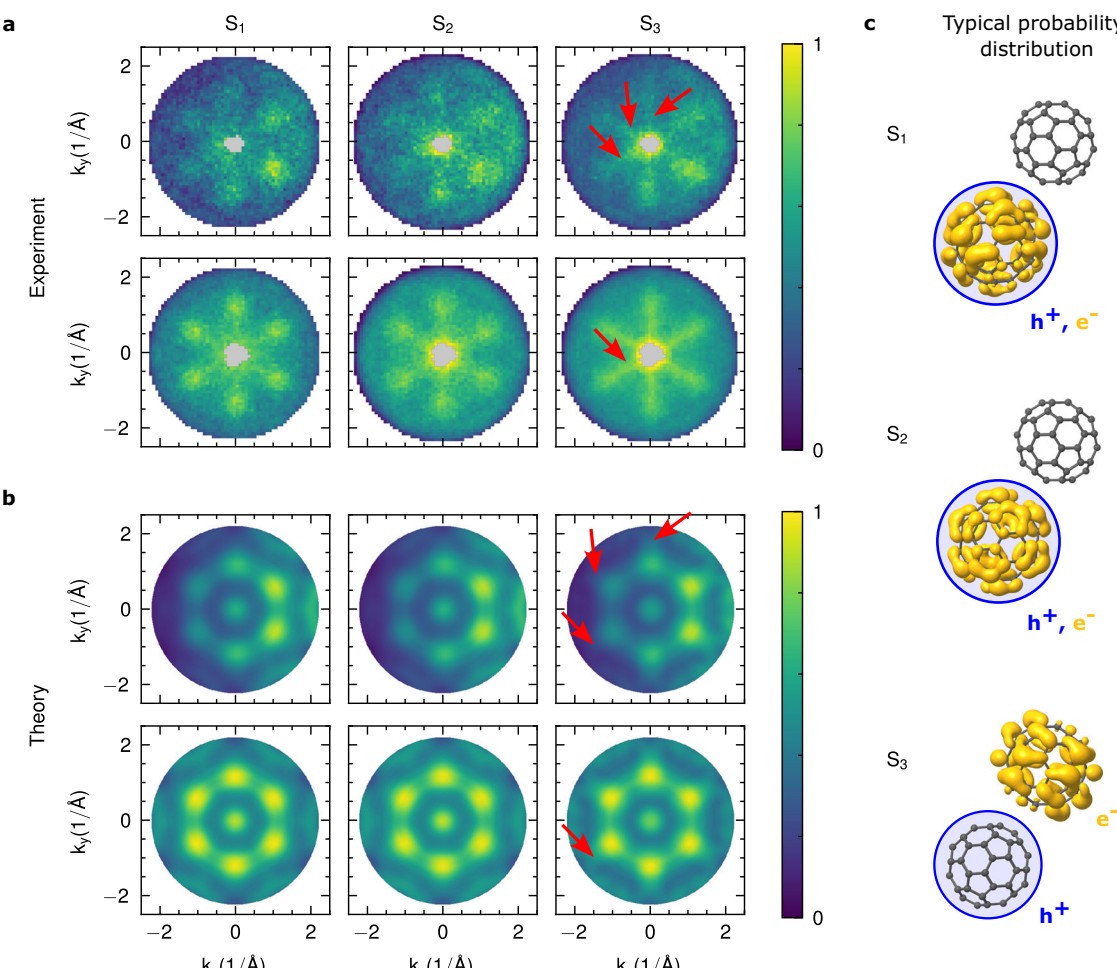

**Fig. 4 | Photoelectron momentum distribution and exciton real space probability distribution of the three exciton bands observed in C$_{60}$. a** Comparison of the experimental momentum maps acquired for the S$_1$, S$_2$ and S$_3$, with the **b** predicted momentum maps retrieved from *GW*+BSE. The top rows show the raw data and the bottom rows 6-fold symmetrized data, respectively. Note that the center of the experimental maps could not be analyzed due to a space-charge-induced background signal in this region (gray area, see Methods). **c** Isosurfaces of the integrated electron probability density (yellow) within the 1-2 dimer for fixed hole positions on the bottom-left molecule (blue circle) of the dimer for the S$_1$, the S$_2$, and the S$_3$ exciton bands.

the eye (Fig. 4b, top right and bottom right, respectively; lineout analysis in Fig. S6). However, we also find that the experimentally observed spoke-like pattern for S$_3$ is different to the S$_3$ momentum structure calculated using the dimer model. An indication towards the cause of this discrepancy is found by considering the electron-hole separation of the excitons making up the S$_3$ band. Here, we find that the positions of the electron and the hole contributions are strongly anticorrelated (Fig. 4c), with the electron confined to the neighboring molecule of the dimer. In fact, the mean electron-hole separation is as large as 7.6 Å, which is close to the core-to-core distance of the C$_{60}$ molecules. Although these theoretical results confirm the previously-reported charge-transfer nature of the S$_3$ excitons[24,25], they also reflect the limitations of the C$_{60}$ dimer approach. Indeed, the dimer represents the minimal model to account for an intermolecular exciton delocalization effect, but it cannot fully account for dispersion effects[42] (see Supplementary Fig. S1), which are required for a quantitative comparison with experimental data. Besides the discrepancy in the S$_3$ momentum map, this could also be an explanation why the S$_2$ in the present work is of Frenkel-like nature, but could have charge-transfer character according to previous studies[24,25]. However, future developments will certainly allow scaling up of the cluster size in the calculation and to include periodic boundary conditions, so that exciton wavefunctions with larger electron-hole separation can be accurately described. Most

importantly, we find that the present dimer *GW*+BSE calculations are clearly suited to elucidate the multiorbital character of the excitons, which is an indispensable prerequisite for the correct interpretation of time- and momentum-resolved data of excitons in organic semiconductors.

In conclusion, we introduced photoemission exciton tomography to unravel the multiorbital electron and hole contributions of entangled excitonic states. In a case study on C$_{60}$, we show how to connect time- and angle-resolved photoelectron spectroscopy data to the wavefunction of fully-interacting exciton states. For the hole component of the exciton, the spectral position of the hole is reflected in the photoelectron kinetic energy distribution, leading to the appearance of multiple peaks in the photoelectron spectrum for a multiorbital exciton. At the same time, the momentum fingerprint provides access to the electron states that make up the exciton. Applying this analysis to the observed exciton photoelectron spectrum of C$_{60}$, we were able to access important key properties including different orbital contributions, the wavefunction localization, and the charge-transfer character. We anticipate that photoemission exciton tomography will contribute to the understanding of exciton dynamics and the harnessing of such particles not only in organic semiconductors, but in general to advanced optoelectronic and photovoltaic devices.

## Methods

### Femtosecond momentum microscopy of $C_{60}$/Cu(111)

We apply full multidimensional time- and angle-resolved photoelectron spectroscopy (tr-ARPES) to a multilayer $C_{60}$ crystal evaporated onto Cu(111), where the film thickness was such that no photoemission signature of the underlying Cu(111) could be observed in our experiment. We verified the sample quality by performing momentum microscopy of the occupied HOMO and HOMO-1 states simultaneously to the measurement of the excited states (see Supplementary Fig. S1). Femtosecond exciton dynamics were induced using ≈100 fs, hν = 2.9 eV or ≈100 fs, hν = 3.6 eV laser pulses derived from the frequency-doubled output of a optical parametric amplifier. The exciton dynamics were probed using our custom photoemission momentum microscope with a 500 kHz ultrafast 26.5 eV extreme ultraviolet (EUV) light source[39] that enables us to map the photoelectron momentum distribution over the full photoemission horizon in a kinetic energy range exceeding 6 eV and an overall time resolution of ≈100 fs (the EUV pulse length is about 20 fs). The pump fluence was set to 90(10) μJ/cm² and 20(5) μJ/cm² for the hν = 2.9 eV and the hν = 3.6 eV measurement, respectively. In both cases, we found that p-polarized light excites the material most efficiently, leading to an excitation density on the order of 1 excitation per 1000 molecules. To prevent the free rotation of $C_{60}$ molecules, we cooled the sample down to ≈80 K[43]. In addition to the resulting long-range periodic ordering of the $C_{60}$ crystal, cooling was also observed to prevent light-induced polymerization.

### Momentum microscopy data preprocessing

In the momentum microscopy experiment, a balance has to be found between sufficiently low pump and probe light intensities to avoid space-charge effects, but also having sufficient intensity for the optical excitation (pump) and reasonably short integration times (probe). For the present experiment, we estimate that the pump and probe pulses each induce less than 50 photoelectrons over the full (≈200 × 150 μm²) footprint of the beam. A 40 μm diameter spatial selection aperture and a low threshold voltage then eliminate most of the low-energy photoelectrons and pass less than 1 photoelectron per pulse to the time-of-flight detector. For our settings, small space-charge effects are present in the data, but these do not lead to strong distortions in the band-structure data and can be easily corrected. Therefore, the first step in the data analysis was to subtract a space-charge-induced delay- and momentum-dependent kinetic energy shift which affects the full data set. For this purpose, the central kinetic energy of the HOMO was determined for normalization. To avoid the influence of the $C_{60}$ crystal band structure[42], we fitted a two-dimensional Lorentzian[44] and shifted the kinetic energy distribution accordingly, leading to the expected overall flat shape of the molecular orbitals in the ARPES data.

Although we use narrow-band multilayer mirrors to select the 11th harmonic at hν = 26.5 eV from our laser-based high-harmonic generation spectrum[39], we observe subtle replicas of the HOMO, HOMO-1, and HOMO-2 states in the unoccupied regime of the spectrum that are caused by photoemission from the 13th harmonic at at hν = 31.2 eV. To quantify these replica signals, we fitted the static reference spectrum above $E - E_H = 1.2$ eV (i.e., in the unoccupied regime of the spectrum) with three Gaussian-shaped peaks for the HOMO replicas and an exponential function to account for residual photoemission intensity in the unoccupied regime that is caused in this spectral region by the much stronger direct 11th harmonic one-photon-photoemission from the HOMO state. After carrying out this fitting routine, we are able to calculate clean 11th harmonic spectra (static and time-resolved) via subtraction of the fitted 13th harmonic HOMO replicas. Note that we only subtract the replica signals, but not the background signal that is caused by one-photon-photoemission with the 11th harmonic from the HOMO state, because this background is time-dependent[24], and needs

**Table 1 | Peak parameters for hν = 2.9 eV excitation, extracted using Eq. (3)**

| Exciton | Kin. Energy (eV) | Bandwidth (FWHM) (eV) |
|---|---|---|
| $S_3$ (t = 0 fs) | 2.768(2) | 0.606(4) |
| $S_2$ | 1.978(2) | 0.406(3) |
| $S_1$ | 1.667(1) | 0.362(2) |

Note that we give the full width at half maximum (FWHM) for the bandwidth.

to be explicitly considered in the fitting procedure. The data shown in Fig. 3 of the main text is processed in the way described above.

### Fitting procedure for the time-resolved data

From replica-free trPES data for hν = 2.9 eV excitation, we determine the amplitude $A_i$, kinetic energy $E_i$, and bandwidth $\Delta E_i$ for the $i^{th}$ exciton signature using a global fitting approach. In particular, we apply the model

$$I(E, t) = \frac{A_{S_3}(t)}{\sqrt{2\pi}\Delta E_{S_3}} \exp\left[(E - E_{S_3}(t))^2/\Delta E_{S_3}^2\right]$$
$$+ \frac{A_{S_2}(t)}{\sqrt{2\pi}\Delta E_{S_2}} \exp\left[(E - E_{S_2})^2/\Delta E_{S_2}^2\right]$$
$$+ \frac{A_{S_1}(t)}{\sqrt{2\pi}\Delta E_{S_1}} \exp\left[(E - E_{S_1})^2/\Delta E_{S_1}^2\right]$$
$$+ A_{bg}(t) \exp\left[-E/\tau\right]. \quad (3)$$

Here, the last term is needed to account for the above-mentioned delay-dependent photoemission intensity that is caused by a transient renormalization of the HOMO state, as found in ref. 24.

The fit results of this model applied to the hν = 2.9 eV excitation and momentum-integrated data are shown in Table 1, and Supplementary Fig. S2a for the time-resolved exciton dynamics.

For the measurement with hν = 3.6 eV excitation, we account for the $S_4$ exciton band by extending the model in Eq. (3) with a set of Gaussian peaks with identical temporal evolution, given by

$$I(E, t) = \ldots + \frac{A_{S_4}(t)}{2\sqrt{2\pi}\Delta E_{S_4}} \left(\exp\left[(E - E_{S_4,\text{upper}})^2/\Delta E_{S_4}^2\right]\right.$$
$$\left. + \exp\left[(E - E_{S_4,\text{lower}})^2/\Delta E_{S_4}^2\right]\right), \quad (4)$$

which follows the same notation as Eq. (3). Here, we set $E_{S_4,\text{upper}}$ to be close to 3.6 eV, and following the GW+BSE calculation we set $E_{S_4,\text{upper}} - E_{S_4,\text{lower}} = 1.4$ eV. Fitting this model to the momentum-integrated hν = 3.6 eV excitation data, we find $E_{S_4,\text{upper}} = 3.59(1)$ eV, and for the FWHM of the $S_4$ we find 0.58(3) eV. The time-resolved amplitudes retrieved using this model are shown in Supplementary Fig. S2b. Furthermore, this analysis was used in Fig. 3 of the main text to subtract the exponential background $A_{bg}(t) \exp\left[-E/\tau\right]$ related to the transient broadening of the HOMO state.

### Fitting procedure for the time- and momentum-resolved data

In order to analyze the time-resolved data also momentum-resolved and thereby retrieve the momentum patterns that are shown in Fig. 4 in the main text, we carry out the fitting routine separately for pixel-resolved energy-distribution curves in the momentum distribution (1 pixel corresponds to ≈0.02 Å⁻²). Despite our efforts to reduce space charge, a residual noise signal remains near the center of the photoemission horizon, leading to a small region where the low signal-to-background ratio does not allow a reliable fit. We therefore exclude this region in our analysis (gray areas in Fig. 4 in the main text). Also, for the momentum-resolved data, the replica HOMO background

signals due to the 13th harmonic amounts to 0, 1 or at most 2 counts in the pixel-resolved (momentum-resolved) energy-distribution curves, and can therefore not be fitted and subtracted accurately as described above for the momentum-integrated data. As such, we need to ignore the HOMO replicas from the 13th harmonic in the momentum-resolved analysis. We avoid overfitting of the model in Eq. (3) by fixing the energy and bandwidth of the peaks in the fitting routine to the parameters given in Table 1. Thus, the set of free parameters in the momentum-resolved fitting procedure is limited to $A_{S3}(k)$, $A_{S2}(k)$, $A_{S1}(k)$ and $A_{bg}(k)$. This approach enables the extraction of reliable momentum distributions also for the partially overlapping energy distributions of the $S_1$ and $S_2$. The $1\sigma$ errors for the full momentum maps are shown in Supplementary Fig. S3.

We note that the overall photoemission intensity of the $S_4$ peak in the hv = 3.6 eV excitation data is comparably low due to the sub-50 fs decay to the lower-energy $S_3$ excitons (see Fig. 3 in main text and Supplementary Fig. S2b). Furthermore, with hv = 3.6 eV excitation, two-photon photoemission with 2 * 3.6 eV = 7.2 eV is sufficient to overcome the work function, so that space-charge effects could only be avoided by considerably reducing the hv = 3.6 eV pump intensity. Therefore, the signal-to-noise ratio in these measurements was not sufficient for a momentum-resolved analysis of the $S_4$ exciton data.

### Calculation of the $C_{60}$ exciton spectrum
The ab initio calculation of the exciton spectrum of the $C_{60}$ film was performed in two steps, using a *GW*+BSE approach. For the static electronic structure, we perform calculations for two unique $C_{60}$ dimers, which have been extracted from the known structure of the molecular film[45] (see Fig. 2c, dimers 1–2 and 1–4 respectively). Starting from Kohn-Sham orbitals and energies of a ground state DFT calculation (6-311G*/PBE0+D3)[46–49] using ORCA 5.0.1[50,51], we employ the Fiesta code[52] to self-consistently correct the molecular energy levels by quasi-particle self-energy calculations with the *GW* approximation. To account for polarization effects beyond the molecular dimer, we embed the dimer cluster in a discrete polarizable model using the MESCal program[53–55]. We found that mimicking 2 layers of the surrounding $C_{60}$ film in such a way resulted in the convergence of the band gap within 0.1 eV with a removal energy from the highest valence level of 6.65 eV. The close agreement of this quasi-particle energy with the experimentally determined work function of 6.5 eV gives us additional confidence in the choice of our embedding environment. The calculated quasi-particle energy levels are shown in Fig. 2a. Here the finite width of the black bars actually arises from multiple energy levels forming bands on the energy axis. We characterize them according to symmetry[56] as HOMO-1, HOMO, LUMO, and LUMO+1 bands, each consisting of 18, 10, 6 and 6 energy levels per dimer, respectively. Note that the HOMO-1 is made up by states from two different irreducible representations of the isolated gas phase $C_{60}$ molecule which are practically forming a single band.

Building upon the *GW* energies, we compute neutral electron-hole excitations by solving the Bethe-Salpeter equation beyond the Tamm-Dancoff approximation (TDA). This yields the excitation energies $\Omega_m$ and the electron-hole coupling coefficients $X_{vc}^{(m)}$ for a series of excitons labeled with $m$. We first analyze the resulting optical absorption spectrum which is shown in Fig. 2c as black solid line. It reveals a prominent absorption band around hv = 3.6 eV that is well-known from gas-phase spectroscopy[57]. Secondly, the dimer calculation reveals a strong optical absorption at hv = 2.8 eV as well as a weakly dipole-allowed transition at hv = 2 eV. Both of these transitions are known to only appear in aggregated phases of $C_{60}$[23], and cannot be understood by considering only a single $C_{60}$ molecule. We refer to the exciton bands around $\Omega$ = 1.9, 2.1, 2.8 and 3.6 eV as $S_1$, $S_2$, $S_3$ and $S_4$, respectively.

In line with our classification of the *GW* energy levels, one can group the composition of the excitons into four categories according

to the contributing quasi-particle energy levels. As visualized in Fig. 2c, this shows that $S_1$ and $S_2$ are almost completely described by HOMO to LUMO transitions. On the other hand, $S_3$ is predicted to have a small contribution of HOMO-1 character, however this contribution is too small to be reliably measured in our experiment. Finally, for $S_4$ we observe a clear and almost equal mixture of HOMO-1 and HOMO contributions, which is also confirmed by our measurements. Contributions from lower lying valence bands are negligibly small in the studied energy window.

### Calculation of the exciton momentum maps
Based on our Kohn-Sham orbitals and BSE excitation coefficients, we calculate theoretical momentum maps for each exciton according to Eq. (2) following the derivation of Kern et al.[19]. Note that for better readability, Eqs. (1) and (2) are given within the TDA, however, can in general be extended to include also de-excitation terms[19]. In the present case, we found the de-excitation contributions to be marginal (below 1%) without affecting the appearance of the momentum maps and our interpretation. The energy conservation term comprises the BSE excitation energies ($\Omega_m$), the *GW* quasi-particle energies for electron removal (i.e. ionization potential $\varepsilon_v$), and the probe energy of hv = 26.5 eV in accordance with our experimental setup. Furthermore, we include an inner potential to correct for the photoemission intensity variation of 3D molecules along the moment vector component perpendicular to the surface. Here, we choose a value of 12.5 eV, which has already been shown to match with experimental $C_{60}$ data[42,58]. (Note that while considering the inner potential of the film is essential to describe the ARPES fingerprint, a variation of the inner potential between 12 and 14 eV indicated no influence on the interpretation of our results).

As we exploited the plane-wave approximation, the calculated photoemission intensity is modulated by the momentum-dependent polarization factor $|Ak|^2$, which we modeled as p-polarized light incoming with 68° to the surface normal according to experiments. To account for the symmetry of the $C_{60}$ film, the momentum maps were 3-fold rotated and mirrored. Finally, application of Eq. (2) provides us with a 3D data set of simulated photoemission intensity as a function of the kinetic energy ($E_{kin}$) and the momentum components $k_x$ and $k_y$ for each individual exciton with excitation energy $\Omega_m$. Analogous to experiment, we referenced the kinetic energy against the energy of the HOMO. The calculated ionization potential was further used to set the photoemission horizon of the theoretical momentum maps. Next, we estimate the population of the different excitonic states and sum up the corresponding 3D photoelectron intensity distributions. Here, we note that the experimental linewidth of the excitonic features is significantly broadened by various external factors, such as inhomogeneity in the sample and a finite energy resolution of the experiment. Therefore, we assume that the different excitonic states within each band are populated equally. For the $S_1$, this includes all excitons from 1.8–2.0 eV, for $S_2$ the 2.0–2.2 eV range, and for $S_3$ 2.7–3.0 eV. Finally, to arrive at the theoretical momentum maps shown in Fig. 4, we integrate the total signal in a wide kinetic energy range centered on the respective exciton band.

## Data availability
The data sets that support the experimental findings of this study are available on GRO.data with the identifier https://doi.org/10.25625/Q7TCIS[59]. The python codes to evaluate the theoretical data can be obtained from the authors upon request.

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

## Acknowledgements

This work was funded by the Deutsche Forschungsgemeinschaft (DFG, German Research Foundation)—432680300/SFB 1456, project B01 and 217133147/SFB 1073, projects B07 and B10. G.S.M.J. acknowledges financial support by the Alexander von Humboldt Foundation. A.W., C.S.K., and P.P acknowledge support from the Austrian Science Fund (FWF) project I 4145 and from the European Research Council (ERC) Synergy Grant, Project ID 101071259. The computational results presented were achieved using the Vienna Scientific Cluster (VSC) and the local high-performance resources of the University of Graz. R.H., M.A., and B.S. acknowledge financial support by the DFG - 268565370/TRR 173, projects B05 and A02. B.S. acknowledges further support by the Dynamics and Topology Center funded by the State of Rhineland-Palatinate. We acknowledge support by the Open Access Publication Funds of the University of Göttingen.

## Author contributions

D.St., M.R., S.S., M.A., B.S., P.P., G.S.M.J. and S.M. conceived the research. W.B., D.Sc. and J.P.B. carried out the time-resolved momentum microscopy experiments. W.B. analyzed the data. W.B. and R.H. prepared the samples. A.W., C.S.K., G.D., X.B. and P.P. performed the calculations and analyzed the theoretical results. All authors discussed the results. G.S.M.J. and S.M. were responsible for the overall project direction and wrote the manuscript with contributions from all co-authors.

## Funding

## Competing interests

The authors declare no competing interests.
