## [Peer Review File · Nature Communications]

REVIEWER COMMENTS

Reviewer #1 (Remarks to the Author):

In this manuscript, Bennecke et al. have employed time-resolved momentum microscopy to map the electron and hole orbital contributions for the excitonic states of the prototypical organic semiconductor C60. The authors present a theoretical framework for the interpretation of such photoemission tomography maps as relevant for the study of excitons, and thereby identify and separate the photoemission signals of states with varying combinations of HOMO(-1) and LUMO(+1,+n) weight. Using these experimentally and theoretically determined momentum space maps, the authors then predict the localization or charge-transfer character of the various excitonic states. The power of this approach to gain simultaneous insight into the dynamics, energetics, and orbital character of the excited states of such organic systems is a promising framework.

My key concern is that the experimental momentum maps are aggressively symmetrized and the authors do not provide much to convince the reader that this heavily processed data is faithful to the raw measurement and is not simply selected to match the theory. Even with this strong symmetrization, the photoemission exciton tomography aspect of the paper is a bit unsatisfying. The S3 exciton signal notably does not really match the corresponding theoretical map, the S2 exciton conclusion of localization is at odds with previous measurements' findings of the charge-transfer character of this state (which were done by the same group, in Ref. 25), and the S4 exciton map is unattainable due to the statistics/long integration time required for the experimental measurement. Taken together, this unfortunately muddles and limits the physical insight intended to be conveyed here.

Still, I do believe that this work constitutes an important step for the burgeoning field of excited-state orbital tomography and for the community's understanding of how to apply these (technically very challenging) measurements to push beyond just demonstration of the technique. To this end, substantiating this work with more clear evidence that the experiment and theory can actually work together effectively to reveal new insight would allow this paper to hold true to its claims of the promise and capabilities of exciton tomography. This paper could be well suited for publication in Nature Communications if the authors can address the following major concerns (and minor comments):

Major comments:

1. As indicated above, my primary concern is that the data here is over-processed by essentially full 6-way symmetrization, which does not inspire confidence that the quality of the experimental data is sufficient for comparison to theory. Naturally the electric field direction of the polarized light will break the symmetry in the image, but this is ignored here. Can the authors provide convincing data that is

unsymmetrized (or at least minimally processed) to demonstrate that their symmetrization approach does not cherry-pick the part(s) of the momentum maps they wish to show for congruence with theory? This would be appropriate to show for the HOMO and HOMO-1 momentum maps as well, where I expect the statistics for the images should be quite good, as a simpler baseline demonstrating how well the theory with this dimer model captures the experiment. I am also particularly puzzled that the same rotation and mirror symmetrization has been applied to the theoretical momentum maps as mentioned in VI. D, even after the effort has been taken to account for the 68 deg AOI p-polarized EUV that should help account for asymmetry in the experimental maps.

2. It appears that in Fig. 4, each of three theoretical maps is the summation of a 200 meV kinetic energy range centered around the band, while the experimental maps energy ranges are given by the global fit model that yields bandwidths much larger than this. How does this impact the agreement (or lack thereof) between the experimental and theoretical maps? Especially given the broad bandwidth of the S3 band from the experimental fit (600 meV) this would seem to be quite important. Can the authors fit the theoretical 4D data set with the same global fit model as the experimental data? Does this yield similar band centers, band widths, and time-resolved amplitudes? It seems that this would yield a more 1:1 comparison of the experiment vs. theory maps. Also, what pump-probe time delay do each of these maps correspond to? Are they essentially static for a given state / kinetic energy except for the amplitude/intensity?

3. It is concerning that the data in Fig. 3 appears to have been taken with a much higher pump fluence for the 2.9 eV excitation than the 3.6 eV excitation. Can the authors justify taking a difference map of these two very different measurements by demonstrating the pump fluence dependence of the data? Does 2.9 eV excitation with 20 $\mu\text{J}/\text{cm}^2$ give the same results as with 90 $\mu\text{J}/\text{cm}^2$? Or can the authors account for this difference in the fluence by a difference in the absorption at each wavelength? Still it appears that the statistics for the energy distribution curves in 3d are also markedly worse for the 3.6 eV data despite the strong calculated absorption peak for this resonance.

4. In the absorption spectrum in Fig. 2c, what is this unmentioned peak at ~ 3.35 eV? There seems to be quite a lot of activity at this energy particularly for H-1 \rightarrow L and H \rightarrow L+1 but I did not see any mention of this either in the main text or in the extended data/methods.

5. It is very interesting in Fig. 6 that the S2 state is more heavily populated following relaxation of initially-pumped S4 than S3. Certainly the presented data supports the conclusion offered in the figure caption that S4 may relax to both states. Can the authors comment on the underlying scientific meaning of this conclusion based on the physical nature of the two states, their orbital character, and their localization? I.e., is it reasonable to expect such a decay pathway, is it congruent with the other experimental findings in the paper, or what can we learn from this observation? And is this conclusion valid given the large difference between the pump fluence used for these two datasets? In my opinion, a

reasonable physical explanation of this also lends validity to the application of this global fit model to the data and would strengthen the rationale for using it.

More minor comments:

1. In general, I found the presentation in both Figs. 1 and 2 to be a bit confusing.

a. While Fig. 1 is meant to be a cartoon, it is confusing that here S2 is demonstrated with these many orbital configurations but in Fig. 2 this is not the case, so perhaps S4 would be a better example choice. The exciton absorbance spectrum is also quite different from the calculated one in Fig. 2 and this incongruence is also confusing. Beyond this, S0-S4 have not been introduced by Fig. 1, so using Ω_1 , Ω_2 , etc. would be clearer, and similarly the meaning of L, L+1 in 1c is undefined.

b. In Fig. 2, I think the individual kinetic energy cartoons across the bottom of 2d is confusing and doesn't really convey how these states should actually appear relative to one another. It would be more illustrative to have one plot showing how all 4 states/5 peaks are expected to line up, and this will provide a better transition for the reader into the data in Fig. 3. Alternatively, a numerical energy axis on each of these 3 cartoon x-axes could clarify this point.

2. I am a bit confused how S3 arises when it is the combination of $H \rightarrow L$ and $H \rightarrow L+1$ although L and L+1 are ≥ 1 eV separated in the energy scale on 2b. From 2c it looks like the weight that is L+1 in S3 is miniscule compared to that of L – is this important? In general I found the introduction/explanation of S1-S4 to be muddled, likely because the notes on the expected charge transfer or delocalization character of each state is 'hidden' until the end when the tomography result is revealed, although there is a considerable amount of past work exploring the nature of these various exciton states in C60.

3. It would be relevant to include both the pump and probe laser polarizations in VI.A. It would likely also be relevant to include some metric of the probe fluence or photoelectrons/pulse if the space-charge effects are so prominent that there is a momentum-dependent kinetic energy shift as seems to be the case here.

4. There is a figure reference missing or typo in line 605 (VI.B.4).

Reviewer #2 (Remarks to the Author):

In the manuscript "Disentangling the multi-orbital contributions of excitons by photoemission exciton tomography", Wiebke Bennecke et al. used pump-probe ARPES to study the exciton tomography of C60 films, and discovered time-, energy- and momentum-resolved change of photoelectron signals.

The research field of exciton physics is one of vibrant activity and the development of advanced experimental techniques for investigating excitons in materials is certainly of importance. However, my primary concern lies in the lack of novelty in this work. The authors claim to have achieved "unprecedented access" to the properties of the entangled exciton state, but unfortunately, there seems to be insufficient evidence to suggest that the techniques used and the results obtained are significantly novel or constitute a groundbreaking contribution to the field.

The technique of time-resolved photoemission momentum microscopy itself is not new and has been used in similar contexts in recent years (Ref.14-18, 40-41). The authors also suggest that they have achieved novel insights into the exciton's properties, including its localization, charge-transfer character, and ultrafast exciton formation and relaxation dynamics. However, again, it is unclear how the reported findings are significantly distinct or advance beyond the current state of knowledge in this area.

In its present form, the paper does not convincingly demonstrate that it meets the journal's criteria for originality and substantial advancement of the field. However, the technical aspects of the work and the clarity of presentation are commendable. The excited-state dynamics of C60 is also interesting. If the authors can address the additional concerns listed below, the paper may be suitable for publication in a more specialized journal.

1. The authors claimed that spatial or real-space properties of the exciton can be obtained. However, the measurement of photoelectron signal generally does not provide the phase information of the exciton wavefunction. As a result, it is challenging to reconstruct real-space exciton wavefunctions by ARPES alone. The authors should clarify this limitation in the revised manuscript.

2. The authors used C60 dimers to model excitonic states in the theory. But this approach largely ignored the dispersion necessary to resolve exciton band. I would suggest the authors to directly performed GW+BSE calculations on periodic C60 systems for an appropriate comparison between the experiment and theory.

Reviewer #3 (Remarks to the Author):

This manuscript reports the application of photoemission momentum microscope to obtain exciton wavefunction tomography in the model system of C60 thin films. Using a combination of GW/BSE calculation and time-resolved momentum imaging of photoelectrons, the authors demonstrated the feasibility of resolving orbital contributions to the three lowest energy exciton states. This was a heroic experiment, e.g., requiring 70 hr integration time and subtle subtraction procedures. This demonstration extends very successful photoemission momentum tomography (e.g., those of HOMOs in the Fig. 5) to

the excited states, namely excitons. While the authors applied the technique to a well known system here to establish feasibility, there is potential of applying this powerful approach to solve important problems. In this regard, the value of this manuscript lies in the method validation. I would recommend publication after the authors have address some the technical questions below.

1) I appreciate the difficulty of the experiment. The authors need to balance avoiding space charge problems with obtaining sufficient signal to noise. This is evident in the signal near $k = 0$. Could the authors comment on how far in momentum space the space charge problem extends to? Have the authors done experiments on excitation or ionization laser power dependences?

2) Fig. 3 aimed to identify the contributions from S4 from excitation at the higher photon energy of 3.6 eV. While the peak at the high kinetic energy end (3.4-4.1 eV) is convincing, the identification of the lower energy feature required the subtraction and less obvious. The overall signal-to-noise ratio at $h\nu = 3.6$ eV is obviously lower than that at 2.9 eV and the authors relied on normalization/time-shifting to show the difference. What is the reason for the lower signal-to-noise ration at $h\nu = 3.6$ eV?

3) The key experimental data in Fig. 4 is of good quality but the comparison with theory is less so. For example, the image from S3 (Fig. 4c) shows a spoke pattern, but the theory gives a more diffused spoke, with a strong ring at intermediate momentum. The authors attributed the discrepancy to the limitation of a dimer model. In fact, the three experimental images (a-c) are not sufficiently different. Each contains a spoke feature, with diminishing intensities from the six spots (ac). This part of the manuscript needs to be improved. Can the calculation be improved to provide more satisfactory agreement? This is important as the authors concluded that S3 contain charge transfer characters.

1 Response to Referee 1

In this manuscript, Bennecke *et al.* have employed time-resolved momentum microscopy to map the electron and hole orbital contributions for the excitonic states of the prototypical organic semiconductor C60. The authors present a theoretical framework for the interpretation of such photoemission tomography maps as relevant for the study of excitons, and thereby identify and separate the photoemission signals of states with varying combinations of HOMO(-1) and LUMO(+1,+n) weight. Using these experimentally and theoretically determined momentum space maps, the authors then predict the localization or charge-transfer character of the various excitonic states. The power of this approach to gain simultaneous insight into the dynamics, energetics, and orbital character of the excited states of such organic systems is a promising framework.

My key concern is that the experimental momentum maps are aggressively symmetrized and the authors do not provide much to convince the reader that this heavily processed data is faithful to the raw measurement and is not simply selected to match the theory. Even with this strong symmetrization, the photoemission exciton tomography aspect of the paper is a bit unsatisfying. The S3 exciton signal notably does not really match the corresponding theoretical map, the S2 exciton conclusion of localization is at odds with previous measurements' findings of the charge-transfer character of this state (which were done by the same group, in Ref. 25), and the S4 exciton map is unattainable due to the statistics/long integration time required for the experimental measurement. Taken together, this unfortunately muddles and limits the physical insight intended to be conveyed here.

Still, I do believe that this work constitutes an important step for the burgeoning field of excited-state orbital tomography and for the community's understanding of how to apply these (technically very challenging) measurements to push beyond just demonstration of the technique. To this end, substantiating this work with more clear evidence that the experiment and theory can actually work together effectively to reveal new insight would allow this paper to hold true to its claims of the promise and capabilities of exciton tomography. This paper could be well suited for publication in Nature Communications if the authors can address the following major concerns (and minor comments):

We thank the reviewer for the critical assessment of our work, and we are pleased that the "paper could be well suited for publication in Nature Communications", if we are able to improve the manuscript with respect to the referee's concerns.

In our revised version, we are confident that we have addressed all the points raised, and we now show unsymmetrized momentum maps. Most importantly, in our revision we recognized that the molecular overlayer structure we had previously inferred from the literature did not perfectly fit to our experimental structure determination. So we recalculated all the theoretical results and ended up with a much better and much more satisfactory agreement between experiment and theory.

We are therefore confident that our manuscript now stands up to its claims and the the reviewer can support its publication in Nature Communications.

Comment 1.1

As indicated above, my primary concern is that the data here is over-processed by essentially full 6-way symmetrization, which does not inspire confidence that the quality of the experimental data is sufficient for comparison to theory. Naturally the electric field direction of the polarized light will break the symmetry in the image, but this is ignored here. Can the authors provide convincing data that is unsymmetrized (or at least minimally processed) to demonstrate that their symmetrization approach does not cherry-pick the part(s) of the momentum maps they wish to show for congruence with theory? This would be appropriate to show for the HOMO and HOMO-1 momentum maps as well, where I expect the statistics for the images should be quite good, as a simpler baseline

demonstrating how well the theory with this dimer model captures the experiment. I am also particularly puzzled that the same rotation and mirror symmetrization has been applied to the theoretical momentum maps as mentioned in VI. D, even after the effort has been taken to account for the 68 deg AOI p-polarized EUV that should help account for asymmetry in the experimental maps.

Answer 1.1: We appreciate the reviewers concern about the symmetrization, and have taken the opportunity to re-evaluate the unsymmetrized data. Figure A1 shows the momentum maps extracted from the unsymmetrized data by the global fitting approach. As pointed out by the reviewer the asymmetry due to the electric field direction of the polarized light is visible in the intensity distribution of the momentum maps (p-polarized light, AOI of 68° along the x-direction). Without symmetrization data quality is reduced and the standard deviation of the fit increases, but the principal features of the excited momentum maps are completely preserved. In particular, the more pronounced star shape pattern of the S_3 remains visible. Furthermore, the expected sixfold symmetry is unambiguously present which justifies the used symmetrization in the paper. Still, echoing the statement that "symmetrization does not inspire confidence", we have chosen to replace the symmetrized momentum maps by their unsymmetrized counterparts.

Figure A1: Amplitude (**a, b, c, d**) and standard deviation (**e, f, g, h**) of the extracted momentum maps without any symmetrization of the S_1 , S_2 , S_3 and the exponential background ($A_{bg}(k)$), respectively. The data are scaled to the mean value inside the circled area.

In response to this question and to Comments from Reviewers 2 and 3, we have revisited the comparison between the experimental and theoretical momentum fingerprints. From the observed crystal band structure in the occupied states (Fig. S1), the orientation of the 2x2 unit cell of the top C_{60} monolayer (w.r.t. the 1x1 structure of the bulk C_{60} crystal at room temperature) can be directly defined. On the other hand, the energy-integrated momentum fingerprint is directly linked to the molecular orbitals and therefore the orientation of the molecules within the unit cell. We can therefore use static photoemission orbital tomography to determine the orientation of the individual C_{60} molecules by comparing the momentum maps to predictions from (ordinary) density functional theory. The results of this analysis are shown in Fig. A2. From our analysis, we can confirm the 2x2 superstructure of the surface layer, but we find that the orientation of the C_{60} molecules is rotated by 90° compared to the orientation that was reported by Wang *et al.* (PRB 63, 085417 (2001)). We have therefore added Fig. A2 to the Supplementary Information, recalculated all theoretical results, and adapted the manuscript accordingly. In addition, our static photoemission orbital tomography serves to benchmark the accuracy of the dimer model compared to the full periodic structure calculation. From the good match between the different theoretical momentum fingerprints in Fig. A2, we conclude that the dimer model is indeed applicable. Most

Figure A2: Comparison of the DFT-calculated momentum maps for a fully periodic monolayer calculation (top row) and for the C_{60} dimer model (middle row) to experiment (bottom row). For the occupied bands, experimental and theoretical momentum maps were retrieved by integrating over the respective energy ranges, while for the DFT LUMO, we show a comparison to the experimental S_1 momentum fingerprint. The orientation of the 2×2 unit cell was matched to the experimentally observed Brillouin zone (see Fig. S1), while the orientation of the molecules was adapted to the observed ARPES momentum fingerprints, i.e., we use static photoemission orbital tomography. Here, the similarity between theory and experiment is most pronounced for the HOMO-2 state, where we observe the strongest contrast in the experiment and no effect of band dispersion. A similar resemblance is seen for the HOMO-1 state when compared with the periodic calculations, while the experimental momentum map of the HOMO does not have clear enough features to be related to the theory. Due to computational complexity, it is not possible at this stage to perform the full BSE calculations on top of the periodic system in an adequate surface (slab) geometry with sufficient amount of vacuum and number of layers. Nevertheless, we can compare the static DFT LUMO and the experimental S_1 momentum map. Same as for the occupied states, the symmetry and general features of the momentum maps are well reproduced. Note that the orientation of the C_{60} molecules in the unit cell was found to be 90° rotated compared to an earlier STM study [Wang et al., 2001]. From the qualitative agreement between the experimental and theoretical momentum maps, we conclude that the dimer model provides an accurate description.

importantly, we find that the experimentally determined structure leads to a much improved and satisfactory match between the experimental and theoretical exciton momentum maps, particularly for the S_3 (see Fig. A3).

Finally, we discuss the validity of the dimer model in the answer to comment 2.3 of Reviewer 2. As explained there, we have also added further information on this to the Supplementary Information.

We have made the following changes to the manuscript:

- Concerning comparison of the occupied orbital fingerprints and concomitantly the determination of the C_{60} crystal structure, we have added Fig. S4 (Fig. A2). Furthermore, we added "see *Supplementary Information, Supplementary Figs. S1, S4 and S5 for an experimental determination of the crystal structure and a convergence analysis of the dimer model*" in lines 90-91.
- As we have found a different C_{60} crystal structure from experiment, we have updated the theoretical results of Figs. 2 and 4 to match this structure. In Fig. 2, we have also made small changes to the caption: dimer 1-4 was changed to 3-4, and the reference to Wang et al., 2001 was removed. (A citation to Wang et al.

Figure A3: Updated version of Fig. 4 in the main article, where we have chosen to show the data without 6-fold symmetrization.

Figure A4: Updated version of Fig. S1 in the main article, where we show the data without 6-fold symmetrization.

is still present in the Supplementary Information).

- As the unsymmetrized data are of sufficient quality for the comparison with theory, we have chosen to replace the experimental momentum maps in Fig. 4 with the unsymmetrized maps. We have also adapted the theoretical momentum maps to reflect the unsymmetrized A·k matrix element.
- Similarly, we have updated Fig. S1 (with Fig. A4) and Fig. S3 (with Fig. A1).

Comment 1.2

It appears that in Fig. 4, each of three theoretical maps is the summation of a 200 meV kinetic energy range centered around the band, while the experimental maps energy ranges are given by the global fit model that yields bandwidths much larger than this. How does this impact the agreement (or lack thereof) between the experimental and theoretical maps? Especially given the broad bandwidth of the S3 band from the experimental fit (600 meV) this would seem to be quite important. Can the authors fit the theoretical 4D data set with the same

global fit model as the experimental data? Does this yield similar band centers, band widths, and time-resolved amplitudes? It seems that this would yield a more 1:1 comparison of the experiment vs. theory maps. Also, what pump-probe time delay do each of these maps correspond to? Are they essentially static for a given state / kinetic energy except for the amplitude/intensity?

Answer 1.2 From the reviewers questions, we realize that it is necessary to clarify a few aspects of our theoretical results. First of all, although our calculations go beyond the ground state that is accessible by conventional DFT methods, our $GW+BSE$ calculation essentially provides a snapshot of the allowed excited states for a fixed pump-probe delay (namely, 0 fs delay). The calculation tells us which e-h pair excited states can exist at this point, and yields the exciton energies and the corresponding electron-hole exciton wavefunctions. However, the calculation does not indicate how a specific excitonic state might relax (other than by light emission). We therefore rely fully on the experimental data to access the timescales of the excitonic relaxation. The momentum maps in Fig. 4 are indeed static in shape except for a global amplitude; and a time-resolved analysis of the momentum-resolved data also supports that the momentum fingerprints do not change with time.

As a consequence of the structure of our theoretical data, it is not possible to apply the exact same fitting method to retrieve the momentum distributions: For this, the time-resolved occupation of each excitonic state would have to be known. Potentially, these time-resolved occupations could be retrieved using a global fitting approach that seeks to match the theoretical momentum maps with the experimentally observed ones, however due to its complexity, such an analysis is beyond the scope of the current article. It is our opinion that the simple, homogeneous occupations that we have estimated in our work better convey the message that it is possible to predict the momentum fingerprints of excitonic states from ab-initio calculations.

Finally, the reviewer correctly notices that the bandwidth of the experimental signatures is significantly broader than the employed bandwidths in the theoretical description. In this aspect, we first note that a similar effect is seen in the occupied states: the experimental signatures are significantly broader than the theoretically predicted dispersion, and we attribute this to both finite linewidth effects as well as local defect-induced shifts. Similarly, we expect that the experimental exciton signatures are broadened compared to the theoretical data, and the observed linewidth is not indicative of the energy spread (temperature) of the excitons. Here, we would like to note that as seen in Fig. 2 the respective theoretical exciton bands are greatly separated in energy such that a broadening of the energy range would not impact the calculated momentum maps.

To clarify how we have calculated the theoretical momentum fingerprints, we have adapted the methods section.

- In lines 470-472, we write "*Finally, application of Eq. 2 provides us with a 3D data set of simulated photoemission intensity as a function of the kinetic energy (E_{kin}) and the momentum components k_x and k_y for each individual exciton with excitation energy Ω_m .*"
- In lines 474-482, we write "*Next, we estimate the population of the different excitonic states and sum up the corresponding 3D photoelectron intensity distributions. Here, we note that the experimental linewidth of the excitonic features is significantly broadened by various external factors, such as inhomogeneity in the sample and a finite energy resolution of the experiment. Therefore, we assume that the different excitonic states within each band are populated equally. For the S_1 , this includes all excitons from 1.8 to 2.0 eV, for S_2 the 2.0 to 2.2 eV range, and for S_3 2.7 to 3.0 eV. Finally, to arrive at the theoretical momentum maps shown in Fig. 4, we integrate the total signal in a wide kinetic energy range centered on the respective exciton band.*"

Comment 1.3

It is concerning that the data in Fig. 3 appears to have been taken with a much higher pump fluence for the 2.9 eV excitation than the 3.6 eV excitation. Can the authors justify taking a difference map of these two very different measurements by demonstrating the pump fluence dependence of the data? Does 2.9 eV excitation with 20 $\mu\text{J}/\text{cm}^2$ give the same results as with 90 $\mu\text{J}/\text{cm}^2$? Or can the authors account for this difference in the fluence by a difference in the absorption at each wavelength? Still it appears that the statistics for the energy distribution

curves in 3d are also markedly worse for the 3.6 eV data despite the strong calculated absorption peak for this resonance.

Answer 1.3: While we agree that a different pump fluence can, in principle, lead to different dynamics, we are confident that the pump fluence does not invalidate the comparison between 3.6 and 2.9 eV excitation. First of all, we note that for both measurements, we achieve only a small exciton density on the order of 1 to 10 excitons per 1000 molecules. We can therefore exclude non-linear effects such as exciton-exciton interactions. Secondly, we note that the total deposited power is low in both cases, such that we can rule out differences in the effective temperature between these measurements.

The reviewer makes a valuable observation about the data quality and absorption cross-section for the S_4 exciton. In order to explain the comparatively low signal for the S_4 exciton, we calculated the integrated EUV photoemission matrix elements, and compared the magnitudes for different excitonic states. Here, we find that the S_4 exciton is predicted to have a significantly lower photoemission cross-section than the lower-lying states. This leads to a weaker photoemission signature, and consequently also to a lower signal-to-noise ratio. Note that a more detailed discussion of the signal strength of the S_4 exciton is given in the answer to comment 2 by reviewer 3.

In response to this comment, we made the following changes:

- We have added the following information to the methods section (lines 335-336): "..., leading to an excitation density on the order of 1 excitation per 1000 molecules."
- To clarify the difference in intensity and photoemission cross section, we have reworded the final paragraph of Section 2C (lines 235-239) to "*The explanation for this effect is two-fold: first, the time-resolved signature suggests a very fast relaxation of the S_4 excitons, with relaxation times well below 50 fs (see Fig. S2). Second, a calculation following Eq. 2 predicts an about threefold reduced photoemission matrix element for the S_4 compared to the S_3 band, explaining the overall weaker signal.*"

Comment 1.4

In the absorption spectrum in Fig. 2c, what is this unmentioned peak at ≈ 3.35 eV? There seems to be quite a lot of activity at this energy particularly for $H-1 \rightarrow L$ and $H \rightarrow L+1$ but I did not see any mention of this either in the main text or in the extended data/methods.

Answer 1.4: Yes, as the reviewer correctly notes there is a significant amount of excitonic states in the energetic region from 3 to 3.5 eV, and, indeed, relaxation from the S_4 to these states should be possible. However, we note that one would also expect a double peak structure in this case, but at about 100 - 600 meV lower energies. In this manner, detection of the double peak structure from these exciton states would also confirm our theoretical expectation.

We thank the referee for this observation and we modified the discussion accordingly:

- In line 215-223, we added: 'We note that relaxation to the band of exciton states between 3.0 and 3.5 eV (cf. Fig. 2c) is also possible. However, these excitons also have contributions from the HOMO and HOMO-1 and are also predicted to lead to two distinct peaks at somewhat lower ≈ 3.4 eV above the HOMO and HOMO-1, thus also confirming our expectations from theory. We therefore need to pinpoint the second $H-1 \rightarrow L$ lower energy contribution from either the S_4 or the exciton band around 3.0-3.5 eV, and we do this by analyzing our data at the earliest time of excitation, i.e. before relaxation to the S_2 dark exciton band occurs, which is degenerate at this photoelectron energy of ≈ 2.2 eV.'

Comment 1.5

It is very interesting in Fig. 6 that the S_2 state is more heavily populated following relaxation of initially-pumped S_4 than S_3 . Certainly the presented data supports the conclusion offered in the figure caption that S_4 may relax

to both states. Can the authors comment on the underlying scientific meaning of this conclusion based on the physical nature of the two states, their orbital character, and their localization? I.e., is it reasonable to expect such a decay pathway, is it congruent with the other experimental findings in the paper, or what can we learn from this observation? And is this conclusion valid given the large difference between the pump fluence used for these two datasets? In my opinion, a reasonable physical explanation of this also lends validity to the application of this global fit model to the data and would strengthen the rationale for using it.

Answer 1.5: We thank the reviewer for observing the difference in S_2 population between the two measurements; indeed we have also noted and thought about this difference. It is possible to propose several mechanisms which might explain this difference, however before doing so we would emphasize that we currently have no ab-initio theoretical description of the exciton relaxation rates. Having said that, we speculate that the observed effect is related to the exciton localization. In particular, we note that the $S_4 - S_3$ relaxation requires both a change of the hole state from HOMO-1 to HOMO and the separation of the electron and hole over two molecules. The $S_4 - S_2$, however, conserves the Frenkel nature of the exciton and does not require a charge separation. This effect may explain a comparably large probability for the $S_4 - S_3$ relaxation and compensate for the larger energy difference between these states.

To reflect these ideas in the manuscript, we have added the following to the caption of Fig. S2: "Here, the shared Frenkel nature of the S_4 and S_2 excitons might contribute to a relatively fast S_4-S_2 scattering rate, while the charge-transfer nature of the S_3 exciton implies that not only an energetic relaxation but also a spatial charge separation is necessary for the S_4-S_3 process."

Minor comments:

Comment 1.6

In general, I found the presentation in both Figs. 1 and 2 to be a bit confusing.

a. While Fig. 1 is meant to be a cartoon, it is confusing that here S_2 is demonstrated with these many orbital configurations but in Fig. 2 this is not the case, so perhaps S_4 would be a better example choice. The exciton absorbance spectrum is also quite different from the calculated one in Fig. 2 and this incongruence is also confusing. Beyond this, S_0-S_4 have not been introduced by Fig. 1, so using Ω_{S1} , Ω_{S2} , etc. would be clearer, and similarly the meaning of L , $L+1$ in 1c is undefined.

b. In Fig. 2, I think the individual kinetic energy cartoons across the bottom of 2d is confusing and doesn't really convey how these states should actually appear relative to one another. It would be more illustrative to have one plot showing how all 4 states/5 peaks are expected to line up, and this will provide a better transition for the reader into the data in Fig. 3. Alternatively, a numerical energy axis on each of these 3 cartoon x-axes could clarify this point.

We would like to thank the reviewer for this very constructive and helpful feedback to improve our figures.

Answer 1.6a: We agree with the reviewer that our initial visualization leads to confusion. Therefore, we have adapted Fig. 1 to represent a more general exciton spectrum. We have also updated the caption to define the notation L and $L+1$.

Answer 1.6b: We agree with the reviewer, and therefore we have added the labels " $E_H + \Omega_{S1,S2}$ ", " $E_H + \Omega_{S3}$ ", " $E_{H-1} + \Omega_{S4}$ " and " $E_H + \Omega_{S4}$ ".

Comment 1.7

I am a bit confused how S_3 arises when it is the combination of $H \rightarrow L$ and $H \rightarrow L+1$ although L and $L+1$ are ≈ 1 eV separated in the energy scale on 2b. From 2c it looks like the weight that is $L+1$ in S_3 is miniscule compared to that of L – is this important? In general I found the introduction/explanation of S_1-S_4 to be muddled, likely because the notes on the expected charge transfer or delocalization character of each state is 'hidden' until the end when the tomography result is revealed, although there is a considerable amount of past work exploring the nature of these various exciton states in C60.

Answer 1.7: Overall, we completely agree that the composition of S_3 out of L and L+1 states being $\geq 1\text{eV}$ apart is rather counter-intuitive. On the other side, the realization that excitons with a particular exciton energy can be composed of coherent sums of orbitals being energetically that far apart is one of the main findings of our work. In this regard, it is important to note that in a *GW/BSE* approach, the exciton wavefunction is expressed in terms of the single-particle excitation energies, that is, using the quasi-particle energy levels (HOMO, LUMO, LUMO+1, ...) of our system. However, their energetic difference serve only as a first guess for the optical excitation energies of the system. Only in the independent particle approximation, that is, in an uncorrelated electron-hole picture, would the energy differences of electron and hole quasi-particle energies correspond to the optical excitation energies. By solving the Bethe-Salpeter equation, we instead explicitly consider correlated electron-hole pairs, leading to the observed – and sometimes maybe surprising – mixing of single-particle contributions.

In order to make this point more clear, we made the following changes:

- In lines 124-128, we revised the final sentence of the corresponding paragraph to "Note that the inclusion of electron-hole correlations has important consequences on the composition of the exciton wave function ψ_m [34,35]. Specifically, S_3 is not only composed of H \rightarrow L transitions but it exhibits also an admixture of H \rightarrow L+1 transitions despite the calculated $\approx 1\text{ eV}$ energy separation of quasi-particle LUMO and LUMO+1 levels."
- In lines 115-117, we added "Notably, S_2 and S_3 were previously found to have charge-transfer character [24, 25], while S_4 stands out due to a fundamentally different wavefunction composition."
- In line 121, we have added the word "weak" when referring to the LUMO+1 contribution.
- We have reworded lines 284-288 to "This correlates with the different wavefunction composition of the S_3 excitons, where the electronic part contains contributions from the LUMO and LUMO+1 orbitals in a coherent sum that is different from S_1 and S_2 ."

Comment 1.8

It would be relevant to include both the pump and probe laser polarizations in VI.A. It would likely also be relevant to include some metric of the probe fluence or photoelectrons/pulse if the space-charge effects are so prominent that there is a momentum-dependent kinetic energy shift as seems to be the case here.

Answer 1.8: With regards to the polarizations, we thank the reviewer for noting this oversight. Both the EUV probe beam and probe beam are p-polarized. We have observed no difference in the dynamics between s and p-polarized pumping, other than an overall reduced absorption for s-polarization.

Concerning the EUV probe fluence, we first emphasize that a measurement of this quantity is rather challenging, and we have no calibrated measurement method to achieve such data. In practice, the EUV fluence is tuned by optimizing HHG conditions and adapting the gas pressure, and the EUV fluence is monitored by counting the total number of photoelectrons that arrive at the detector. From our measurements, we estimate that during measurement, less than 50 photoelectrons are created per probe pulse over the full EUV spot ($\approx 200 \times 150 \mu\text{m}^2$). The pump pulse induces a similar number of photoelectrons per pulse. However, as the number of detected electrons in comparison to the emitted electrons depends dramatically on the properties and settings of the spectrometer, we emphasize that these numbers provide little value when comparing different experiments.

We have made the following changes to the manuscript to clarify these points:

- In line 199, we added "p-polarized excitation".
- In lines 334-336, we added "In both cases, we found that p-polarized light excites the material most efficiently".
- In lines 342-346, we added "For the present experiment, we estimate a that the pump and probe pulses each induce less than 50 photoelectrons over the full ($\approx 200 \times 150 \mu\text{m}^2$) footprint of the beam. A 40 μm diameter

spatial selection aperture and a low threshold voltage then eliminate most of the low-energy photoelectrons and pass less than 1 photoelectron per pulse to the time-of-flight detector."

Comment 1.9

There is a figure reference missing or typo in line 605 (VI.B.4).

Answer 1.9: We have corrected this and deleted the missing reference.

2 Response to Referee 2

In the manuscript "Disentangling the multi-orbital contributions of excitons by photoemission exciton tomography", Wiebke Bennecke *et al.* used pump-probe ARPES to study the exciton tomography of C₆₀ films, and discovered time-, energy- and momentum-resolved change of photoelectron signals.

The research field of exciton physics is one of vibrant activity and the development of advanced experimental techniques for investigating excitons in materials is certainly of importance. However, my primary concern lies in the lack of novelty in this work. The authors claim to have achieved "unprecedented access" to the properties of the entangled exciton state, but unfortunately, there seems to be insufficient evidence to suggest that the techniques used and the results obtained are significantly novel or constitute a groundbreaking contribution to the field.

The technique of time-resolved photoemission momentum microscopy itself is not new and has been used in similar contexts in recent years (Ref. 14-18, 40-41). The authors also suggest that they have achieved novel insights into the exciton's properties, including its localization, charge-transfer character, and ultrafast exciton formation and relaxation dynamics. However, again, it is unclear how the reported findings are significantly distinct or advance beyond the current state of knowledge in this area. In its present form, the paper does not convincingly demonstrate that it meets the journal's criteria for originality and substantial advancement of the field. However, the technical aspects of the work and the clarity of presentation are commendable. The excited-state dynamics of C₆₀ is also interesting. If the authors can address the additional concerns listed below, the paper may be suitable for publication in a more specialized journal.

We thank the reviewer for the examination of our work, and appreciate the positive words regarding the technical aspects of our work and the clarity of its presentation. We want to note that in response to the Reviewers, we have refined the C₆₀ structure in the theoretical description, and have thereby achieved an even more convincing comparison with the experimental data (see Comment 1.1 for a complete explanation).

In response to the reviewer's concerns about novelty, however, we would like to emphasize that we present a completely novel analytical framework that allows to directly benchmark state-of-the-art, fully-interacting exciton calculations with time-resolved momentum microscopy, and that thereby enables to study the entangled nature of excitons using photoemission orbital tomography (POT). We are convinced that our manuscript contains sufficient original and important content that significantly advances the field and our understanding of exciton properties in organic semiconductors, even without our analysis of C₆₀ exciton localization, charge transfer character and excited state dynamics (see also our answer 2.1).

Specifically, there is no other work that shows clearly, from experiment and theory, how the exciton bands are composed of multi-orbital contributions. In particular, we want to emphasize the realization that excitons ψ_m that have a single well-defined energy Ω_m can counter-intuitively be composed of orbitals with *GW* energies that may differ by more than 1 eV, which we show in theory and confirm in experiment. Moreover, there is no other work that realizes that the photoelectron momentum distribution is solely determined by the electron (multi-)orbital contribution to the exciton, and there is no other work that shows that clearly how the exciton's photoelectron energy is determined by the exciton's hole position in the band structure. While the latter point has of course

been realized previously and may seem trivial at first glance, the situation drastically changes and gets more complicated for charge-transfer excitons in hybrid structures, where the exciton's hole and electron are found in different layers. In such systems, the concept of the "binding energy" of the exciton is not well defined anymore, and our work is of utmost importance to understand the experimental exciton signatures and their interpretation.

We are fully convinced that our work is fundamental to understand exciton properties in organic semiconductors, and for all future APRES studies on exciton dynamics, in particular with respect to the interpretation of orbital contributions, and therewith localization and charge-transfer character.

Comment 2.1

The technique of time-resolved photoemission momentum microscopy itself is not new and has been used in similar contexts in recent years (Ref.14-18, 41-42). The authors also suggest that they have achieved novel insights into the exciton's properties, including its localization, charge-transfer character, and ultrafast exciton formation and relaxation dynamics. However, again, it is unclear how the reported findings are significantly distinct or advance beyond the current state of knowledge in this area.

Answer 2.1: We respectfully disagree with the reviewer and argue here that our findings represent a significant step beyond the state of the art in time-resolved momentum microscopy (trMM), as we extend the powerful methods of photoemission orbital tomography to excitons in organic semiconductors. As demonstrated by Refs. 14-18, 41-42 and several more articles, the development of trMM has enabled a highly valuable multidimensional view on the photoelectron spectra of many materials and devices, and indeed already in its current state trMM data can be evaluated to yield a physical understanding. However, in particular for the molecular systems that we study here, it is important to investigate precisely what information can be extracted from the multidimensional data. This is analogous to the original development of POT from occupied HOMO levels, where it was found that a Fourier transform can directly link the molecular orbital (or Dyson orbital, to be precise) to the photoemission data. However, as shown in our article, this relation cannot be directly extended to excitonic states, which is a key problem for the development and interpretation of all future *time resolved* orbital imaging tomography studies, which intrinsically start with a short pulse optical excitation of excitons in the material. In our article, we validate for the first time a theoretical model which can be applied to excitons in organic semiconductors, and it is this model which now allows to study the properties of the excitonic wavefunction. This is a highly important step forward for the study of organic semiconductors using time-resolved photoemission and especially momentum microscopy, as in contrast to 2D and 3D semiconductors, the angle-resolved photoemission signature of an organic semiconductor cannot be analysed easily in terms of valence and conduction bands.

It is true that exciton properties such as localization and charge transfer can be (and have also been) analyzed by other, more indirect measurements. For instance, the charge-transfer nature of the S_3 and S_2 was inferred from a broadening of the valence band structure. By probing the excitonic wavefunction itself, we open up a new avenue for the study of these properties that will be beneficial not just for the C_{60} case but for exciton dynamics in organic semiconductor materials or heterostructures in general.

To further highlight the novelty of our work, we have revised the introduction:

- We have rearranged lines 54 - 62 to better highlight the challenge of accessing the multiorbital contributions of an exciton. The text now reads: *"Access to this orbital picture of the excitonic wavefunction is highly valuable [9], because imaging of the full entangled state would give direct access to exciton properties such as localization and charge-transfer character. Indeed, this information is particularly critical in the case of organic semiconductors, where it is well-known that such multiorbital correlated quasiparticles dominate the energy landscape [10,11]. However, it must be emphasized that conventional optical spectroscopy methods, including absorption and fluorescence spectroscopy, only provide access to the exciton energy Ω_m , and do not provide any information about the multiorbital contributions ($\phi_v^* \chi_c$) of the exciton (cf. Fig. 1a and 1b)."*
- We have also rewritten the last part of the introduction (lines 70-72) to include *"In this article, we experimentally introduce photoemission exciton tomography (Fig. 1c) and use it for the first time to characterize the correlated excitonic electron-hole state in an organic semiconductor."*

Comment 2.2

The authors claimed that spatial or real-space properties of the exciton can be obtained. However, the measurement of photoelectron signal generally does not provide the phase information of the exciton wavefunction. As a result, it is challenging to reconstruct real-space exciton wavefunctions by ARPES alone. The authors should clarify this limitation in the revised manuscript.

Answer 2.2: Generally in photoemission orbital tomography, there are two approaches to achieve access to the real-space wavefunction: either by directly analyzing the ARPES pattern using a phase retrieval algorithm, or by comparing the ARPES pattern to ab-initio calculations of the wavefunctions using a suitable model of photoemission. It is correct that in this case, the former method does not apply, however our results show here that the latter approach is certainly useful: it allows us to characterize the correlated two-particle exciton wavefunction.

In order to clarify this point, we have added a statement to the caption of Fig. 1: "*A full momentum- and energy- resolved measurement of the photoelectron spectrum therefore provides an ideal starting point for a comparison to ab-initio calculations of the excitonic wavefunction, and thereby provides access to the spatial properties of the exciton.*"

Comment 2.3

The authors used C₆₀ dimers to model excitonic states in the theory. But this approach largely ignored the dispersion necessary to resolve exciton band. I would suggest the authors to directly performed *GW*+BSE calculations on periodic C₆₀ systems for an appropriate comparison between the experiment and theory.

Answer 2.3: The reviewer is of course completely right that a full *GW*+BSE calculation of the periodic structure, in a proper surface geometry with sufficient vacuum and number of layers to mimic the semi-infinite bulk would be preferable. However, given the current state of art and computational power, such calculations were not feasible. The embedded dimer model represents the first step from an isolated gas phase molecule towards the crystal, and is able to capture the charge-transfer effects that have been reported before and that are confirmed by our study. In our answer to Comment 1.1 and the attached Fig. A2, we have demonstrated that the dimer model allows an accurate description of the orbital nature of the C₆₀ crystal.

In order to also validate the exciton spectra from the dimer model, we have done the following:

- We have compared *GW*+BSE calculations of single-molecule, dimer and trimer systems. Here, the single-molecule model cannot predict any charge-transfer states, while the dimer and trimer models do predict such excitons and give a good match with the experimental exciton spectrum. The calculations for the trimer model do not show significant differences compared to the dimer model (see Fig. A5b). However, the increased complexity and computational cost of the trimer calculation implies that full, verified convergence of the calculations for all trimer combinations in the 4-molecule unit cell is currently not feasible.
- Due to the unscreened dipole moment of charge-transfer excitons, the energy alignment of these states is highly sensitive to the dielectric environment. For the dimer model, we have therefore implemented an embedding of the system in a C₆₀ multilayer. As shown in Fig. A5a, by comparing the exciton spectra for embedding in 0, 1, 2, and 3 layers of C₆₀ molecules, this procedure allows us to accurately predict the excitation energy of the charge-transfer excitons (specifically, S₃).

With these observations in mind, we conclude that, at present, the embedded dimer model ultimately leads to more accurate and reliable results.

- As indicated in the answer to Comment 1.1, we have added Fig. S4 (Fig. A2) to address the determination of the C₆₀ crystal structure and compare the periodic and dimer-level calculations. Furthermore, we added "*see Supplementary Information, Supplementary Figs. S1, S4 and S5 for an experimental determination of the crystal structure and a convergence analysis of the dimer model*" in lines 90-91.
- To explain that the dimer model in a proper dielectric (screening) environment is sufficiently accurate for our purposes, we have added Fig. S5 (Fig. A5) to the manuscript.

Figure A5: Convergence test of the embedded $GW+BSE$ calculation with respect to the cluster size and embedding level. a) From the top to bottom panel, full $GW+BSE$ calculations were performed for a C_{60} dimer in increasing embedding conditions. Exciton energies are indicated by the black bars on the horizontal axes. The insets show a cut through the considered 3D multilayer systems. We distinguish charge-transfer (red dots) and Frenkel excitons by their e-h distance. As the embedding changes the dielectric environment and therefore the dielectric screening, charge-transfer excitons are most affected by the procedure. This can be observed, as the band of charge-transfer excitons shifts from 3.4 eV to 2.8 eV when going from 0 to 3-layer embedding. As the exciton spectrum changes only negligibly going from 2 to 3 layers of embedding, we conclude that 2-layer embedding provides an accurate description of the excitons. b) Comparison of the exciton spectra of a C_{60} dimer and a representative C_{60} trimer, at identical levels of embedding. The larger size of the trimer allows for more charge-transfer excitons, but the overall shape of the absorption spectrum does not differ significantly from that of the dimer. As the computational cost of the trimer is significantly larger, higher embedding is currently not feasible. We therefore conclude that, at present, the dimer model leads to the most accurate and reliable results.

3 Response to Referee 3

This manuscript reports the application of photoemission momentum microscope to obtain exciton wavefunction tomography in the model system of C₆₀ thin films. Using a combination of GW/BSE calculation and time-resolved momentum imaging of photoelectrons, the authors demonstrated the feasibility of resolving orbital contributions to the three lowest energy exciton states. This was a heroic experiment, e.g., requiring 70 hr integration time and subtle subtraction procedures. This demonstration extends very successful photoemission momentum tomography (e.g., those of HOMOs in the Fig. 5) to the excited states, namely excitons. While the authors applied the technique to a well known system here to establish feasibility, there is potential of applying this powerful approach to solve important problems. In this regard, the value of this manuscript lies in the method validation. I would recommend publication after the authors have addressed some the technical questions below.

We thank the reviewer for the kind words and positive evaluation.

Comment 3.1

I appreciate the difficulty of the experiment. The authors need to balance avoiding space charge problems with obtaining sufficient signal to noise. This is evident in the signal near $k = 0$. Could the authors comment on how far in momentum space the space charge problem extends to? Have the authors done experiments on excitation or ionization laser power dependencies?

Answer 3.1: The reviewer is correct that a balance between detection rate and space charge effects is necessary. In practice, this means that the influence of space charge on each data set (at each unique pump-probe delay) is characterized and corrected. In the present case, this lead to a momentum-dependent shift of the photoelectrons to earlier detection time (apparently higher energy). This shift is strongest in the center (normal emission) and weaker close to the photoemission horizon, but it affects the full data set. As explained in the Methods section, we have extracted and corrected this shift based on the signal of the occupied molecular orbitals.

A second, related effect is the appearance of an increased background signal close to the normal emission. This energy-independent background signal prevents an accurate fit of the excited state signal, and therefore we have chosen to exclude these pixels from the published momentum maps. The extent of this background signal is visible from the gray pixels in Fig. 4 and Supplementary Fig. S3.

With regards to fluence-dependent measurements: Both the pump and the probe beams are limited in intensity to avoid excessive space charge effects. In this regime, the photoionization is purely perturbative, with at most one photoionization event per 1.000.000 molecules per laser shot. In comparison, the pump beam interacts more strongly with the C₆₀ sample. By comparing the total photoemission yield of the HOMO to that of the excited states, we estimate that for 2.9 eV excitation, where we achieve the highest exciton density, the exciton density is on the order of 1 to 10 excitons per 1000 molecules. We therefore conclude that we are in a low-power regime where neither nonlinear optical processes nor exciton-exciton interactions play a role.

In order to explain some of the details about the balance between signal strength and space-charge distortion, we made some modifications to the methods section.

- In line 342-346, we added "*For the present experiment, we estimate that the pump and probe pulses each induce less than 50 photoelectrons over the full footprint of the beam. A 40 μm diameter spatial selection aperture and a low threshold voltage then eliminate most of the low-energy photoelectrons and pass less than 1 photoelectron per shot to the time-of-flight detector.*"
- With regard to the effect of space-charge on the experimental data, we added "*which affects the full data set*" in line 350.

Comment 3.2

Fig. 3 aimed to identify the contributions from S4 from excitation at the higher photon energy of 3.6 eV. While the peak at the high kinetic energy end (3.4-4.1 eV) is convincing, the identification of the lower energy feature

required the subtraction and less obvious. The overall signal-to-noise ratio at $h\nu = 3.6$ eV is obviously lower than that at 2.9 eV and the authors relied on normalization/time-shifting to show the difference. What is the reason for the lower signal-to-noise ratio at $h\nu = 3.6$ eV?

Answer 3.2: The lower signal-to-noise ratio for 3.6 eV excitation is due to a number of reasons, including lower pump fluence, short lifetime, a slightly off-resonant spectrum of the pump pulse, and finally a weaker EUV photoemission matrix element. From these reasons, the lower pump fluence is necessary to avoid space charge effects: as the photon energy is high enough to drive 2-photon photoemission processes, significantly more unwanted photoelectrons are generated, which contribute to the overall space charge. Further, we accumulate less excitons in the S_4 states for two reasons: the short S_4 lifetime leads to the formation of S_3 and S_2 excitons already within the duration of the pump pulse, and since the pump pulse spectrum (at 3.6 eV) is shifted slightly with respect to the calculated absorption maximum at 3.8 eV, we also generate less initial excitations (note that the optical parametric amplifier was not able to provide a stable output at 3.8 eV). The lower signal directly leads to a lower signal-to-noise ratio. Finally, our theoretical calculations predict that the S_4 exciton band has a threefold lower photoemission matrix element, leading to an altogether weaker ARPES signature.

Nevertheless, we observe a good signal for the S_3 (as well as S_2 and S_1) for both photon energies, and it is this signal that we have used to align the time axes and to fix the normalization.

To clarify this, we have reworded the final paragraph of Section 2C (lines 235-239) to "*The explanation for this effect is two-fold: first, the time-resolved signature suggests a very fast relaxation of the S_4 excitons, with relaxation times well below 50 fs (see Fig. S2). Second, a calculation following Eq. 2 predicts an about threefold reduced photoemission matrix element for the S_4 compared to the S_3 band, explaining the overall weaker signal.*"

Comment 3.3

The key experimental data in Fig. 4 is of good quality but the comparison with theory is less so. For example, the image from S_3 (Fig. 4c) shows a spoke pattern, but the theory gives a more diffused spoke, with a strong ring at intermediate momentum. The authors attributed the discrepancy to the limitation of a dimer model. In fact, the three experimental images (a-c) are not sufficiently different. Each contains a spoke feature, with diminishing intensities from the six spots (a→c). This part of the manuscript needs to be improved. Can the calculation be improved to provide more satisfactory agreement? This is important as the authors concluded that S_3 contain charge transfer characters.

Answer 3.3: In response to this comment and comments of Reviewers 1 and 2, we revisited the comparison of the experimental and theoretical momentum fingerprints for the occupied and the excited molecular orbitals. As detailed in the answer to Comment 1.1 and Fig. A2, we have now used static photoemission orbital tomography to determine the crystal structure, and found a rotation of the C_{60} molecules in the 2×2 unit cell with respect to literature [Wang et al., 2001]. We have adapted all figures to this updated crystal structure. Crucially, while this improvement changed the S_1 and S_2 momentum maps little, the S_3 momentum map was more affected. From the updated Fig. 4 (reproduced in Fig. A3), we find a much improved agreement between experiment and theory for the S_3 band.

As detailed in the answer to Comment 1.1, the C_{60} crystal structure has been revised, and all theoretical momentum fingerprints have been recalculated and adapted accordingly.

REVIEWER COMMENTS

Reviewer #1 (Remarks to the Author):

In their revised manuscript, the authors have made a number of revisions that have markedly improved the clarity of the manuscript and the presentation of the results. The improved clarity of the dimer model and the addition of Fig S5 is particularly helpful, and I appreciate the addition of the unsymmetrized data and the HOMO comparisons in addition to the calculation of the S4 photoemission matrix element to clarify the comparison with the S3 signal. These latter two points are particularly important for the demonstrative nature of this work and generally for the adoption of (exciton) photoemission tomography as a more widespread technique and how such data should be analyzed and interpreted in the community. I still have a key remaining concern regarding the now revised experimental and theoretical momentum maps presented in Fig. 4 (see below); if the authors can remedy this important weak point, I support the publication of this work.

As mentioned, I appreciate seeing the unsymmetrized data in Fig. 4 and the added HOMO comparisons in Fig. S4. I am inclined to agree with the authors that the experimental momentum map for the S3 signal may have a slightly different 'spoke' or star-shaped pattern that may be absent in the S1 and S2 experimental maps, but my concern from looking at the images a, b, and c is that the distinction between S3 and any of these other maps is extremely subtle. Similarly, although the updated theoretical description has indeed objectively improved the agreement between the S3 experimental and theoretical momentum maps, now all three theoretical maps in d, e, and f also appear identical or nearly so. This significantly weakens the authors' argument here about the importance of the momentum-resolved measurement as the insight regarding multiorbital contributions is already apparent from the momentum-integrated data in Fig. 3 without tomography, and it is hard to believe that we have really learned something here about the character of these excitons from the tomography when the states all look the same but the conclusion of their character is different. To this end, can the authors provide some other additional plots or way to allow the reader to visualize differences in the images among the experimental maps and among the theoretical maps in Fig. 4 to demonstrate if/that these various exciton maps are distinct in some way? Perhaps with careful normalization of the experimental data this could be done with difference images, or otherwise some plots of lineouts to pick out any subtle differences. This would be important to show comparisons between the experimental images, between the theoretical images, and then comparing the experiment with theory. I recognize the significant difficulty of both the experimental measurement and the theoretical calculations performed here, but without this improved analysis/presentation, the last few pages of the manuscript comprising the tomography section are unfortunately quite weak.

The authors have generally addressed and answered my other questions and concerns.

One minor point, I should clarify in comment/answer 1.2 I was asking what experimental pump-probe time delay (or range) each of the experimental momentum maps presented in Fig. 4 correspond to, i.e., are these images summed up over some range of delays from -100 to 200 fs (or however long the signal persists, since the momentum fingerprints do not change with time as noted in the authors' response) or were they just integrated for a long time at one particular pump-probe delay (and if so, which one(s)).

Reviewer #2 (Remarks to the Author):

In the reply and revised manuscript, the authors have provided a detailed response to the concerns raised during the initial review process. After careful consideration of the authors' revisions and responses, I believe the manuscript has been significantly improved and now justifies publication in Nature Communications.

On the novelty aspect, the authors have argued that "our work is fundamental to understand exciton properties in organic semiconductors, and for all future APRES studies on exciton dynamics, in particular with respect to the interpretation of orbital contributions". I believe this statement correctly reflects the contribution of the work, considering other published ARPES studies of inorganic materials.

The authors have also addressed the concerns regarding their theoretical model. While acknowledging the limitations of their embedded dimer model, they have provided a justified argument for its use over more complex methodology, such as GW+BSE calculations on periodic C60 systems, due to computational constraints. They have further strengthened their argument by demonstrating that their dimer model, with refinements in the theoretical description, provides more insights and achieves a convincing comparison with the experimental data.

Regarding the responses to Referee 3's comments, I believe that authors have provided reasonable justification on the three technical questions raised.

On comment 3.1, the authors argued that the space charge problem has been corrected, and the fluence does not cause nonlinear effects. On comment 3.2, the authors postulated several factors that might have caused the low signal-to-noise ratio at a particular frequency. On comment 3.3, the authors have conducted further calculations (in concert with the response to referee 1 and 2) that improves the match between experiments and theory.

1 Response to Referee 1

In their revised manuscript, the authors have made a number of revisions that have markedly improved the clarity of the manuscript and the presentation of the results. The improved clarity of the dimer model and the addition of Fig S5 is particularly helpful, and I appreciate the addition of the unsymmetrized data and the HOMO comparisons in addition to the calculation of the S4 photoemission matrix element to clarify the comparison with the S3 signal. These latter two points are particularly important for the demonstrative nature of this work and generally for the adoption of (exciton) photoemission tomography as a more widespread technique and how such data should be analyzed and interpreted in the community. I still have a key remaining concern regarding the now revised experimental and theoretical momentum maps presented in Fig. 4 (see below); if the authors can remedy this important weak point, I support the publication of this work.

Answer 1.0: We thank the reviewer for the positive review and the chance to improve the discussion of the momentum maps. We are convinced that our revised manuscript remedies this last weak point and can now be published in Nature Communications.

Comment 1.1

As mentioned, I appreciate seeing the unsymmetrized data in Fig. 4 and the added HOMO comparisons in Fig. S4. I am inclined to agree with the authors that the experimental momentum map for the S3 signal may have a slightly different ‘spoke’ or star-shaped pattern that may be absent in the S1 and S2 experimental maps, but my concern from looking at the images a, b, and c is that the distinction between S3 and any of these other maps is extremely subtle.

Answer 1.1: We agree with the referee that the difference between the S1 and S2 vs. the S3 is subtle in the current representation, and we appreciate to have the chance to improve this part of the manuscript. First, we added arrows to the momentum maps in Fig. 4 as a guide to the eye (see below for updated version of the Figure) and considerably improved our discussion on this point (lines 288-302 in main manuscript).

Second, we are convinced that it is also helpful to show the symmetrized data. In our recent talks, experts from the photoemission orbital tomography community were quite surprised that we do only show the raw data, but not the symmetrized versions. We therefore now include the raw data and the symmetrized versions in the main manuscript in the revised Fig. 4 (see below).

Third, we carried out a lineout analysis, as suggested by the referee, and added these data in Supplementary Fig. S6 (see below). In the lineout analysis of the experimental data, once again, the qualitative and quantitative differences of the lobe structure of S1 and S2 vs. the spoke structure of S3 can clearly and unambiguously be determined. In theory, the difference is less clear, but still visible in the plots and the analysis. We note that we discussed the reason for the different structures in experiment and theory for the S3 exciton band in the main manuscript in lines 302-319.

Changes to the manuscript 1.1:

- We revised the discussion about the S3 exciton band in lines 288-302 in the main manuscript:
“Compared to the S_1 and S_2 exciton bands, the S_3 band is not only composed of $H \rightarrow L$ transitions, but also has a minor contribution of $H \rightarrow L+1$ transitions (cf. Fig. 2c). We therefore expect that the S_3 momentum map cannot be identical to the S_1 and S_2 momentum maps, but must show a signature of the $H \rightarrow L+1$ contribution in addition to a possibly different coherent sum of all involved $H \rightarrow L$ transitions. Indeed, a closer look at the experimental data shows a more spoke-like structure for S_3 , which is marked with red arrows for three of the six spokes in the raw data as a guide to the eye (Fig. 4a, top right). An analysis over all different orientations, i.e. effectively

symmetrizing the data, makes the spoke-structure even better visible (Fig. 4a, bottom right, and Supplementary Fig. S6 for momentum lineouts). Hence, our experimental data clearly confirms the different character of the S_3 exciton band in comparison to S_1 and S_2 . Looking at the theoretical data, we also find differences between the nearly identical S_1 and S_2 momentum maps (Fig. 4b) in comparison to the S_3 momentum map. Once again, the differences are marked with red arrows as a guide to the eye (Fig. 4b, top right and bottom right, respectively; lineout analysis in Fig. S6). However, we also find that the experimentally observed spoke-like pattern for S_3 is different to the S_3 momentum structure calculated using the dimer model. ...”

- We revised Fig. 4 to better show the differences between the momentum maps, and adapted the references to this figure in the text accordingly.

FIG. 4. **a** Comparison of the experimental momentum maps acquired for the three exciton bands observed in C_{60} with the **b** predicted momentum maps retrieved from $GW+BSE$. The top rows show the raw data and the bottom rows 6-fold symmetrized data, respectively. Note that the center of the experimental maps could not be analyzed due to a space-charge-induced background signal in this region (gray area, see Methods). **c** Isosurfaces of the integrated electron probability density (yellow) within the 1-2 dimer for fixed hole positions on the bottom-left molecule (blue circle) of the dimer for the S_1 , the S_2 , and the S_3 exciton bands.

- We added the lineout analysis of the experimental and theoretical data in the Supplementary Information as Fig. S6:

FIG. S6. **a,b** Comparison of the experimentally observed lobe-structure of the S_1 and S_2 exciton bands in contrast to the spoke-structure of the S_3 exciton band. The main differences are indicated by black arrows in **a**. The photoemission intensity profiles **b** were taken along the lobe and spoke direction as indicated with the colored lines in the symmetrized momentum maps of the S_1 , S_2 , and S_3 exciton bands in **a** (note that the momentum maps have been rotated by 90° with respect to the main text). The lineouts in the waterfall plot **b** clearly show the lobe structure of S_1 and S_2 as intensity peaks at about $\approx 1.25 \text{ \AA}^{-1}$. In contrast, the lineout of the S_3 exciton band does not exhibit a peak, but a nearly uniform intensity distribution between $\approx 0.5 - 1.5 \text{ \AA}^{-1}$, clearly indicating the spoke-structure of S_3 . **c,d** A similar analysis was performed for the calculated momentum maps. Here, the S_3 exciton band shows a less pronounced, but still visibly distinct structure in comparison to the S_1 and S_2 , as indicated by black arrows in the symmetrized momentum maps **c**. In the lineouts in **d**, the difference is visible by reduced intensity in the $\approx 0.0 - 0.6 \text{ \AA}^{-1}$ momentum range and an additional shoulder in the $\approx 1.3 - 2.2 \text{ \AA}^{-1}$ momentum range. However, the theoretical model does not reproduce the same spoke-like structure as observed in experiment, as is explained in the main text.

Comment 1.2

Similarly, although the updated theoretical description has indeed objectively improved the agreement between the S3 experimental and theoretical momentum maps, now all three theoretical maps in d, e, and f also appear identical or nearly so. This significantly weakens the authors' argument here about the importance of the momentum-resolved measurement as the insight regarding multiorbital contributions is already apparent from the momentum-integrated data in Fig. 3 without tomography, and it is hard to believe that we have really learned something here about the character of these excitons from the tomography when the states all look the same but the conclusion of their character is different.

Answer 1.2: As in the case of the experimental data, the representation of the theoretical data was not ideal, and we improved this accordingly. In the revised version, the difference between S1 and S2 vs. S3 is now indicated by arrows in the momentum maps shown in Fig. 4b, and the lineout analysis has been added to the Supplementary Information in Fig. S6. In addition, we note that the addition of H→L+1 transitions for the S3 exciton band requires that the momentum map looks different in comparison to S1 and S2. However, we note that the amount of H→L+1 transitions in comparison to H→L transitions is low (see Fig. 2c in main manuscript), so that similarity between all three S1, S2, and S3 momentum maps must still be expected. We recognize that the discussion of these points was rather weak, and we revised the text accordingly in lines 302-319 in the main manuscript.

Changes to the manuscript 1.2: Please see our changes in response to Comment 1.1.

Comment 1.3

To this end, can the authors provide some other additional plots or way to allow the reader to visualize differences in the images among the experimental maps and among the theoretical maps in Fig. 4 to demonstrate if/those these various exciton maps are distinct in some way? Perhaps with careful normalization of the experimental data this could be done with difference images, or otherwise some plots of lineouts to pick out any subtle differences.

Answer 1.3: We thank the referee for the suggestion to show momentum lineouts, which we added as Supplementary Fig. S6 (see our Answer to Comment 1.1). Figs. S6a,b show clearly and unambiguously qualitative and quantitative differences between the lobe-structure of S₁ and S₂ vs. the spoke-structure of the S₃ exciton band in experiment. We note that an analysis of all lobe/spoke momentum directions inevitably leads to a symmetrization, in whichever way such an analysis is carried out, and we still think that symmetrization of the data is fully justified for the given molecular film structure. For comparison, we also added the lineouts of the symmetrized theoretical momentum maps (Fig. S6c,d) which nicely reproduce the main features and similarity between the S₁ and S₂ momentum maps. The theoretical S₃ momentum map shows subtle differences compared to the S₁ and S₂, which are for example visible by a shoulder at $\approx 1.8 \text{ \AA}^{-1}$ indicated by black arrows in the momentum maps (Fig. S6c), and which are also found in the momentum lineouts (Fig. S6d). However, as discussed in the main text, an agreement of the S₃ momentum map with experiment is still missing.

Changes to the manuscript 1.3: We added Supplementary Fig. S6, see our Answer to Comment 1.1. The experimental lineouts of the S₁ and S₂ in Fig. S6b show that the lobes are centered at $k_{\parallel} \approx 1.25 \text{ \AA}^{-1}$. We therefore updated the manuscript in line 266 accordingly.

Comment 1.4

This would be important to show comparisons between the experimental images, between the theoretical images, and then comparing the experiment with theory. I recognize the significant difficulty of both the experimental measurement and the theoretical calculations performed here, but without this improved analysis/presentation, the last few pages of the manuscript comprising the tomography section are unfortunately quite weak.

Answer 1.4: We agree that the former version of this analysis was quite weak, and are happy that we had the chance to improve this part of the manuscript accordingly. We are convinced that the new version of the manuscript with the improved discussion and Figures clearly illustrates the similarities and differences in the momentum maps, and therefore meets all requirements for publication in Nature Communications.

Comment 1.5

The authors have generally addressed and answered my other questions and concerns.

One minor point, I should clarify in comment/answer 1.2 I was asking what experimental pump-probe time delay (or range) each of the experimental momentum maps presented in Fig. 4 correspond to, i.e., are these images summed up over some range of delays from -100 to 200 fs (or however long the signal persists, since the momentum fingerprints do not change with time as noted in the authors' response) or were they just integrated for a long time at one particular pump-probe delay (and if so, which one(s)).

Answer 1.5: We thank the reviewer for the clarification of this question. The momentum maps were integrated over all measured time-steps between -200 fs and 15 ps.

Changes to the manuscript 1.5: In response to this comment, we revised the text in line 253-256: *"We note that the collection of the S_1 , S_2 , and S_3 momentum maps already required integration times of up to 70 hours and a summation of the data over all measured time-steps from -200 fs to 15 ps (see Methods), so that a measurement of the comparatively low-intensity S_4 feature when excited with $h\nu = 3.6$ eV has not yet proved feasible."*

Response to Referee 2

In the reply and revised manuscript, the authors have provided a detailed response to the concerns raised during the initial review process. After careful consideration of the authors' revisions and responses, I believe the manuscript has been significantly improved and now justifies publication in Nature Communications.

On the novelty aspect, the authors have argued that "our work is fundamental to understand exciton properties in organic semiconductors, and for all future APRES studies on exciton dynamics, in particular with respect to the interpretation of orbital contributions". I believe this statement correctly reflects the contribution of the work, considering other published ARPES studies of inorganic materials.

The authors have also addressed the concerns regarding their theoretical model. While acknowledging the limitations of their embedded dimer model, they have provided a justified argument for its use over more complex methodology, such as $GW+BSE$ calculations on periodic C_{60} systems, due to computational constraints. They have further strengthened their argument by demonstrating that their dimer model, with refinements in the theoretical description, provides more insights and achieves a convincing comparison with the experimental data.

Regarding the responses to Referee 3's comments, I believe that authors have provided reasonable justification on the three technical questions raised. On comment 3.1, the authors argued that the space charge problem has been corrected, and the fluence does not cause nonlinear effects. On comment 3.2, the authors postulated several factors that might have caused the low signal-to-noise ratio at a particular frequency. On comment 3.3, the authors have conducted further calculations (in concert with the response to referee 1 and 2) that improves the match between experiments and theory.

Answer 2.0: We thank the reviewer for the positive review.

REVIEWERS' COMMENTS

Reviewer #1 (Remarks to the Author):

The authors have fully addressed all of my comments. I agree that it is reasonable to present the data in Fig. 4 with both the raw and symmetrized data for visualization; this is helpful to show that the symmetrized data is indeed faithful to the raw measurement. The revised description and explanation of the features of the momentum maps and experiment/theory agreement in the text on pages 12 and 13 is also good. The addition of lineouts across the images in Fig S6 is also helpful to more clearly visualize the spoke vs. lobe structure and the shoulder in the S3 theory map.

Again I commend the authors on this obviously very challenging combined experimental and theoretical study.

This work should be published in Nature Communications now as-is.